# Regulation of the terminal maturation of iNKT cells by mediator complex subunit 23

Yu Xu[1], Yang Sun[1], Hao Shen[1], Yuling Dai[1], Haifeng Liu[1], Ronghong Li[1], Hongdao Zhang[1], Ligang Wu[1], Xiaoyan Zhu[1] & Xiaolong Liu [1,2]

Invariant natural killer T cells (iNKT cells) are a specific subset of T cells that recognize glycolipid antigens and upon activation rapidly exert effector functions. This unique function is established during iNKT cell development; the detailed mechanisms of this process, however, remain to be elucidated. Here the authors show that deletion of the mediator subunit Med23 in CD4+CD8+ double positive (DP) thymocytes completely blocks iNKT cell development at stage 2. This dysregulation is accompanied by a bias in the expression of genes related to the regulation of transcription and metabolism, and functional impairment of the cells including the loss of NK cell characteristics, reduced ability to secrete cytokines and attenuated recruitment capacity upon activation. Moreover, *Med23*-deficient iNKT cells exhibit impaired anti-tumor activity. Our study identifies Med23 as an essential transcriptional regulator that controls iNKT cell differentiation and terminal maturation.

[1] State Key Laboratory of Cell Biology, CAS Center for Excellence in Molecular Cell Science, Shanghai Institute of Biochemistry and Cell Biology, Chinese Academy of Sciences, University of Chinese Academy of Sciences, Shanghai 200031, China. [2] School of Life Science and Technology, ShanghaiTech University, Shanghai 200031, China. Correspondence and requests for materials should be addressed to X.L. (email: liux@sibs.ac.cn)

nvariant natural killer T (iNKT) cells are a distinct T cell lineage that expresses an invariant Vα14-Jα18 TCR in mice (Vα24-Jα18 in humans) combined with a limited TCRβ repertoire and surface markers that are typically associated with NK cells. iNKT cells recognize glycolipid antigens presented by the MHC class I-like molecule CD1d and are a functional bridge between the initial innate immune response and subsequent adaptive response[1]. Upon activation, iNKT cells play a protective or detrimental role in infectious and autoimmune diseases and mediate tumor surveillance by rapidly secreting copious levels of cytokines such as interferon-γ (IFN-γ) and interleukin 4 (IL-4) and activating diverse immune cells[2–5]. The chemically synthesized glycolipid α-galactosylceramide (α-GalCer) was originally isolated from a marine sponge and is a ligand for iNKT cells[6]. α-GalCer rapidly activates iNKT cells and exhibits potent anti-tumor activity against various types of tumor cells in mouse models of cancer such as the B16F10 melanoma lung metastasis mouse model[7,8].

iNKT cells are mostly generated in the thymus from CD4$^+$CD8$^+$ double-positive (DP) thymocytes, similar to the generation of conventional T cells. There are four distinct stages of iNKT cell development based on CD24, CD44 and NK1.1 expression on TCRαβ$^+$ T cells that react with the CD1d tetramer loaded with the α-GalCer analog PBS57 (CD1d-PBS57)[1,9–12]. The earliest iNKT cell precursors (stage 0) are derived from DP thymocytes and are defined as CD24$^{high}$CD44$^{lo}$NK1.1$^{lo}$. Subsequently, stage 0 cells downregulate the B cell differentiation marker CD24 and enter into a highly proliferative phase (stage 1; CD24$^{lo}$CD44$^{lo}$NK1.1$^{lo}$). Stage 1 cells further upregulate expression of the activation and memory marker CD44 to become CD24$^{lo}$CD44$^{hi}$NK1.1$^{lo}$ iNKT cells (stage 2) and then develop and differentiate to become fully mature iNKT cells (stage 3; CD24$^{lo}$CD44$^{hi}$NK1.1$^{hi}$). This stage is accompanied by the expression of NK cell surface receptors such as NK1.1, CD94, and NKG2D[13]. This final transition requires TCR signaling mediated by CD1d and is often a point at which a block occurs during iNKT cell development[14]. Recently, based on distinctive functions and transcription factor expression, a novel classification of iNKT cells into NKT1, NKT2 and NKT17 cells was proposed, similar to the classification of effector T helper cells. NKT1 cells express T-bet and secrete IFN-γ primarily at stage 3; NKT2 cells express GATA3 and secrete IL-4 primarily at stage 2; and NKT17 cells express RORγt and produce interleukin 17 (IL-17) primarily at stage 2. These three subsets of iNKT cells are defined by intracellular staining for the transcription factors PLZF, T-bet, and RORγt[15,16]. However, the intrinsic relationship between the lineage diversification model and the linear stage model remains to be investigated.

The mediator, an evolutionarily conserved high-molecular-mass complex composed of more than 20 distinct subunits, plays an important role in the RNA polymerase II general transcriptional machinery and functions directly through RNA polymerase II to integrate and transduce regulatory information from enhancers to promoters performing various activities in transcription[17–19]. One of its subunits, Med23 (also known as Sur2), was originally identified as a target of the viral oncoprotein E1A and responds to a highly specialized signaling pathway[20–22]. Hence, Med23 plays distinct roles in diverse biological processes including carcinogenesis, DNA repair, and bone development[23–25]. Recently, we reported that Med23 contributes to setting the conventional T cell activation threshold by promoting the expression of negative regulators of T cell activation and prevents autoimmunity[26].

Here, we find that loss of Med23 leads to decreased iNKT cell numbers and a complete block at stage 2 during iNKT cell development. To determine how Med23 regulates this process, we first compare the transcriptional and functional features of stage 2 and stage 3 iNKT cells from wild type (WT) mice and then investigate how these features change in the absence of Med23. We find that stage 3 iNKT cells, which express distinct transcriptomes, are able to upregulate surface makers typically associated with NK cells and rapidly secrete cytokines and chemokines upon stimulation compared with stage 2 iNKT cells. However, Med23-deficient iNKT cells fail to do so, even less than WT stage 2 iNKT cells. In particular, the anti-tumor activity of Med23-deficient iNKT cells is significantly attenuated. Overall, we identify Med23 as a critical regulator of the terminal maturation of iNKT cells.

## Results

**Med23 deficiency blocks iNKT cell development at stage 2.** To study the roles of Med23 in conventional T cells, we generated Med23 conditional knockout (Med23$^{-/-}$) mice (designated KO mice) driven by a CD4 promoter in which Med23 is efficiently and specifically deleted by the DP stage of T cell development[26]. Because most iNKT cells arise from DP thymocytes[9–12], we further investigated whether Med23 regulates iNKT cell development. We determined the frequency and absolute number of iNKT cells from WT littermates and Med23$^{-/-}$ mice by CD1d-PBS57 and TCRβ staining. The frequency and cellularity of iNKT cells in the thymi, spleens, and livers of Med23$^{-/-}$ mice were significantly decreased compared with those in the WT controls (Fig. 1a, b and Supplementary Fig. 1a, b), which indicated that Med23 plays important roles in iNKT cell development. iNKT cell development requires an interaction between the semi-invariant TCR and CD1d and is highly dependent on SLAM signaling[1]. Our data indicated that Med23 deficiency in DP thymocytes impacted neither the rearrangement of TCR Vα14-Jα18 and CD1d expression (Supplementary Fig. 2a, c) nor the expression or transcription of the SLAM family of proteins (Slamf1, Slamf6, Fyn and Sap) (Supplementary Fig. 2a, b).

These phenotypes inspired us to further examine the stage at which Med23 deficiency inhibits iNKT cell development. Although stage 0 iNKT cells did not appear to be impacted, Med23-deficient iNKT cells in the thymus, spleen, and liver failed to express NK1.1 to complete final maturation (Fig. 1c, d and Supplementary Fig. 1a, b). There was a significant increase in the percentage of iNKT cells at stage 2 and a severe decrease in the percentage of iNKT cells at stage 3, which indicated that the transition from stage 2 to stage 3 in iNKT cell development was severely affected, and the development of iNKT cells was completely blocked at stage 2 in Med23-deficient mice. Nevertheless, this block did not increase the number of splenic and liver stage 2 cells (Fig. 1e, f). To determine whether the lack of stage 3 cells was attributable to impaired NK1.1 expression in the absence of Med23, we bred conditional Med23$^{fl/fl}$ mice (with a mixed genetic background of C57BL/6:Sv129) with the Vav1-Cre strain, generating a Med23 deletion in the hematopoietic system. The expression of NK1.1 in NK cells was similar to that in WT controls (Supplementary Fig. 3), suggesting that Med23 did not directly regulate NK1.1 expression.

We further examined whether the blocked development of iNKT cells in Med23$^{-/-}$ mice was due to a mechanism intrinsic to the cells by transferring a 1:1 mixture of bone marrow cells from WT (CD45.1$^+$) and Med23$^{-/-}$ (CD45.2$^+$) mice into irradiated Rag2 knockout recipient mice and analyzing iNKT cell development 2 months later. Analysis of these chimeras showed that Med23$^{-/-}$ iNKT cells did not reconstitute efficiently, and their development was blocked at stage 2, whereas WT iNKT cells in the same recipients developed normally (Fig. 1g–j). These data indicated a cell-intrinsic role for Med23 in controlling iNKT cell development.

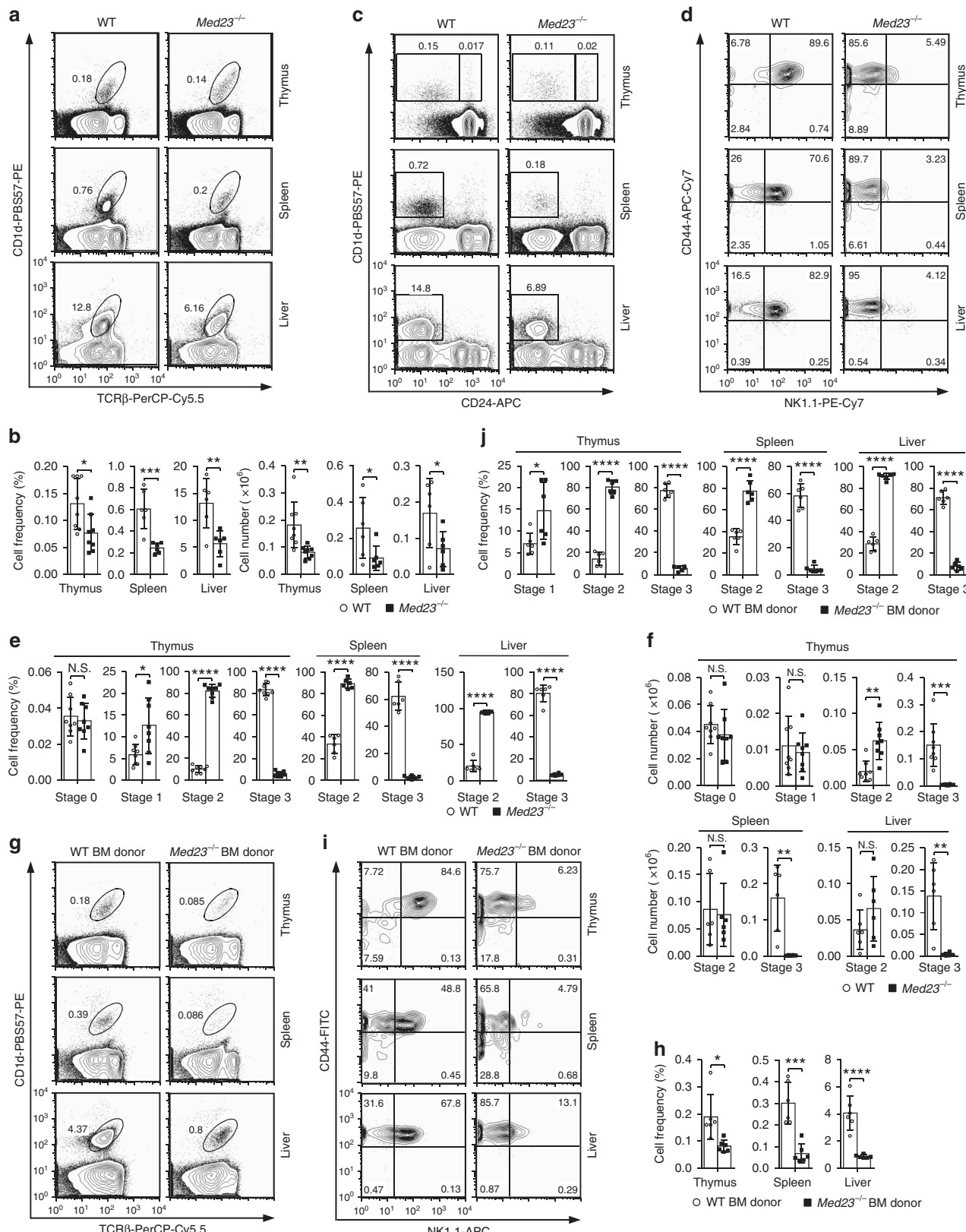

**Loss of Med23 impairs iNKT cell functional differentiation**. To determine whether Med23 deficiency affects iNKT cell function, we stained iNKT cells intracellularly with antibodies against PLZF, T-bet, and RORγt. Compared with WT mice, $Med23^{-/-}$ mice had a higher frequency of NKT2 cells and fewer NKT1 cells and NKT17 cells. In thymic iNKT cells, we observed a similar tendency of the absolute number (Fig. 2a–c and Supplementary Fig. 1c).

**Fig. 1** Deletion of *Med23* blocks stage 2 to stage 3 iNKT cell development. **a** Flow cytometric analysis of TCRβ$^{int}$CD1d-PBS57$^+$ cells in the thymi, spleens, and livers of five- to eight-week-old WT and *Med23*$^{-/-}$ mice. **b** The percentage and number of iNKT cells in the thymi, spleens, and livers of WT and *Med23*$^{-/-}$ mice (thymus, $n = 8$; spleen and liver, $n = 6$). **c, d** Dot plots showing the iNKT cell developmental stages of five- to eight-week-old WT and *Med23*$^{-/-}$ mice. Stage 0 cells were gated as CD24$^{high}$tetramer$^+$ in the thymus (**c**). Gated CD24$^{low}$tetramer$^+$ cells in the thymi, spleens, and livers were analyzed further for CD44 and NK1.1 expression (**d**). **e, f** The frequency (**e**) and number (**f**) of iNKT cells at stage 0, stage 1, stage 2 and stage 3 in the thymi as well as stage 2 and stage 3 in the spleens and livers of five- to eight-week-old WT and *Med23*$^{-/-}$ mice (thymus, $n = 8$; spleen and liver, $n = 6$). **g** Flow cytometric analysis of WT (CD45.1$^+$) and *Med23*$^{-/-}$ (CD45.2$^+$) iNKT cells from the thymi, spleens, and livers of Rag2 knockout recipient mice. **h** The frequency of WT (CD45.1$^+$) and *Med23*$^{-/-}$ (CD45.2$^+$) iNKT cells in the thymi, spleens, and livers of Rag2 knockout recipient mice ($n = 6$). **i** Flow cytometric analysis of CD44 and NK1.1 expression in WT (CD45.1$^+$) and *Med23*$^{-/-}$ (CD45.2$^+$) CD24$^{low}$tetramer$^+$ cells in the thymi, spleens, and livers of Rag2 knockout recipient mice. **j** The frequency of WT (CD45.1$^+$) and *Med23*$^{-/-}$ (CD45.2$^+$) iNKT cells at stage 1, stage 2, and stage 3 in the thymi and stage 2 and stage 3 in the spleens and livers of Rag2 knockout recipient mice ($n = 6$). The data are presented as the mean ± s.d. For all panels: *$P < 0.05$; **$P < 0.01$; ***$P < 0.001$; ****$P < 0.0001$ by Student's *t*-test; N.S. no significance. All data are representative of or combined from at least three independent experiments

Considering the block in iNKT cell development in *Med23*$^{-/-}$ mice, we further surveyed the functional differentiation of stage 2 iNKT cells. *Med23*$^{-/-}$ stage 2 cells had similar frequencies of NKT1 and NKT2 cells and a dramatically lower frequency of NKT17 cells compared with WT control cells (Fig. 2d–i). The frequency of NKT17 cells was lowest among thymic, splenic, and liver WT stage 2 iNKT cells (Fig. 2g–i), suggesting that Med23 did not impact the main functional subsets in stage 2 cells but regulated the generation of NKT17 cells. This differentiation bias was probably a result of higher ThPOK expression in *Med23*$^{-/-}$ stage 2 iNKT cells (Supplementary Fig. 4), consistent with previously published data[27,28]. Moreover, NKT2 cells, which are the primary population of stage 2 cells, include the population which does not produce IL-4 in the steady state and can develop into NKT1 cells in thymi and spleens[15], and no stage 3 NKT1 cells are present in *Med23*-deficient mice. We therefore concluded that the functional differentiation of iNKT cells is impaired in the absence of Med23.

**Med23 controls iNKT cell development partially by c-Jun.** Previous study have shown that interleukin 15 (IL-15) plays an important role in the transition from stage 2 to stage 3 and regulates iNKT cell survival[12,29,30]. Our data indicated that Med23 deficiency did not impair either IL-15 receptor α (IL-15Rα) expression or the levels of apoptosis in stage 2 iNKT cells (Supplementary Fig. 5a-c). Considering that the expression of Med23 gradually increased during the stage 1 to stage 3 transition (Fig. 3a) and that Med23 deficiency completely blocked the transition from stage 2 to stage 3 in *Med23*$^{-/-}$ iNKT cells (Fig. 1d–f), we speculated that Med23 plays a more important role in the transition from stage 2 to stage 3. Hence, we further investigated how Med23 regulated this process. To this end, we compared the transcription-related gene expression in WT stage 2 and stage 3 cells and found that each cell type had distinct characteristics, suggesting that transcriptional regulation was involved in this final transition. Notably, this transition upregulated a series of transcriptional factors associated with Med23, like *Fosl2*, a member of the AP-1 family that is dysregulated when Med23 is mutated in humans;[22] *Foxo4*, a member of the FOXO family that has existing functional interactions with Med23;[31] and *Runx2*, whose transcriptional activity is modulated by Med23 during osteogenesis[24] (Fig. 3b). To study whether Med23 mediates the final transition from stage 2 to stage 3 via these genes, we used *Fosl2* as a template and measured gene expression, including that of AP-1 transcription factors. We observed diverse gene expression between WT stage 2 and stage 3 iNKT cells (Fig. 3c). Moreover, c-Jun, a critical component of AP-1 combined with c-Fos, exhibited decreased expression in stage 2 *Med23*$^{-/-}$ iNKT cells compared with that in WT stage 2 and stage 3 cells (Fig. 3d). We retrovirally expressed c-Jun in *Med23*-deficient

iNKT cells and observed that its ectopic expression partially restored the stage 2 to stage 3 transition of *Med23*$^{-/-}$ iNKT cells (Fig. 3e, f).

Previous studies from other labs investigating the regulation of the transition of iNKT cells have focused on transcription factors and metabolism regulators. We also analyzed the expression of transcriptional and metabolic regulators in stage 2 WT and *Med23*$^{-/-}$ iNKT cells, including *Tbx21*[32], *Egr2*[33], *Med1*[34], *Tsc1*[35], and *Pten*[36]. Stage 2 *Med23*$^{-/-}$ iNKT cells had aberrantly high *Med1*, *Tsc1*, and *Pten* expression compared with that observed in stage 2 WT iNKT cells (Fig. 3g), indicating that Med23 influenced the transcription of certain important regulators in the transition from stage 2 to stage 3. To further confirm our conclusion, we compared the transcriptome of *Med23*$^{-/-}$ stage 2 cells with that of WT controls. Pathway-enrichment analysis of differentially expressed genes revealed that some genes related to chromosome assembly had different expression levels in WT and *Med23*$^{-/-}$ stage 2 iNKT cells (Supplementary Fig. 6). Overall, the expression bias of a series of regulators may give rise to a block in the development of *Med23*$^{-/-}$ iNKT cells, and we demonstrated that this block was partially caused by the disruption of AP-1 activity.

**The functional characteristics of stage 2 and stage 3 cells.** The loss of Med23 led to a block in the development of iNKT cells at stage 2, which encouraged us to investigate how Med23 regulates the functional maturation of iNKT cells from stage 2 to stage 3. First, we compared the biological signatures of stage 2 and stage 3 cells from WT mice because stage 3 cells were absent in *Med23*$^{-/-}$ mice. Pathway-enrichment analysis of differentially expressed genes revealed that many categories related to the immune process exhibited dramatic changes (Fig. 4a). These results prompted us to explore the functional differences between stage 2 and stage 3 cells. Similar to a previous study[37], stage 3 cells had increased levels of not only activating but also inhibiting NK cell surface receptors (Supplementary Fig. 7a). However, *Med23*-deficient stage 2 iNKT cells, which did not exhibit NK cell characteristics, failed to express functional receptors of NK cells such as NK1.1, NKG2D, CD94, and Ly-49G2[38–41] (Supplementary Fig. 7b).

We analyzed the genes encoding cytokine activity that were distinctive between stage 2 and stage 3 cells (Fig. 4b). The expression of *Fasl*, which induces cell apoptosis with the Fas receptor and is important for the killing role of cytotoxic T cells, was higher in stage 3 cells than in stage 2 cells[42]. Stage 3 cells also had higher expression of *Ifng*, which participates in immune responses against viruses, intracellular bacteria and tumors[43], and *Tnfsf11*, which enhances the capacity of dendritic cells to stimulate naïve T cell proliferation[44]. By contrast, stage 2 cells expressed *Il13*, whose aberrant secretion is associated with the pathogenesis of allergic disorders[45], and *Tgfb3*, which regulates

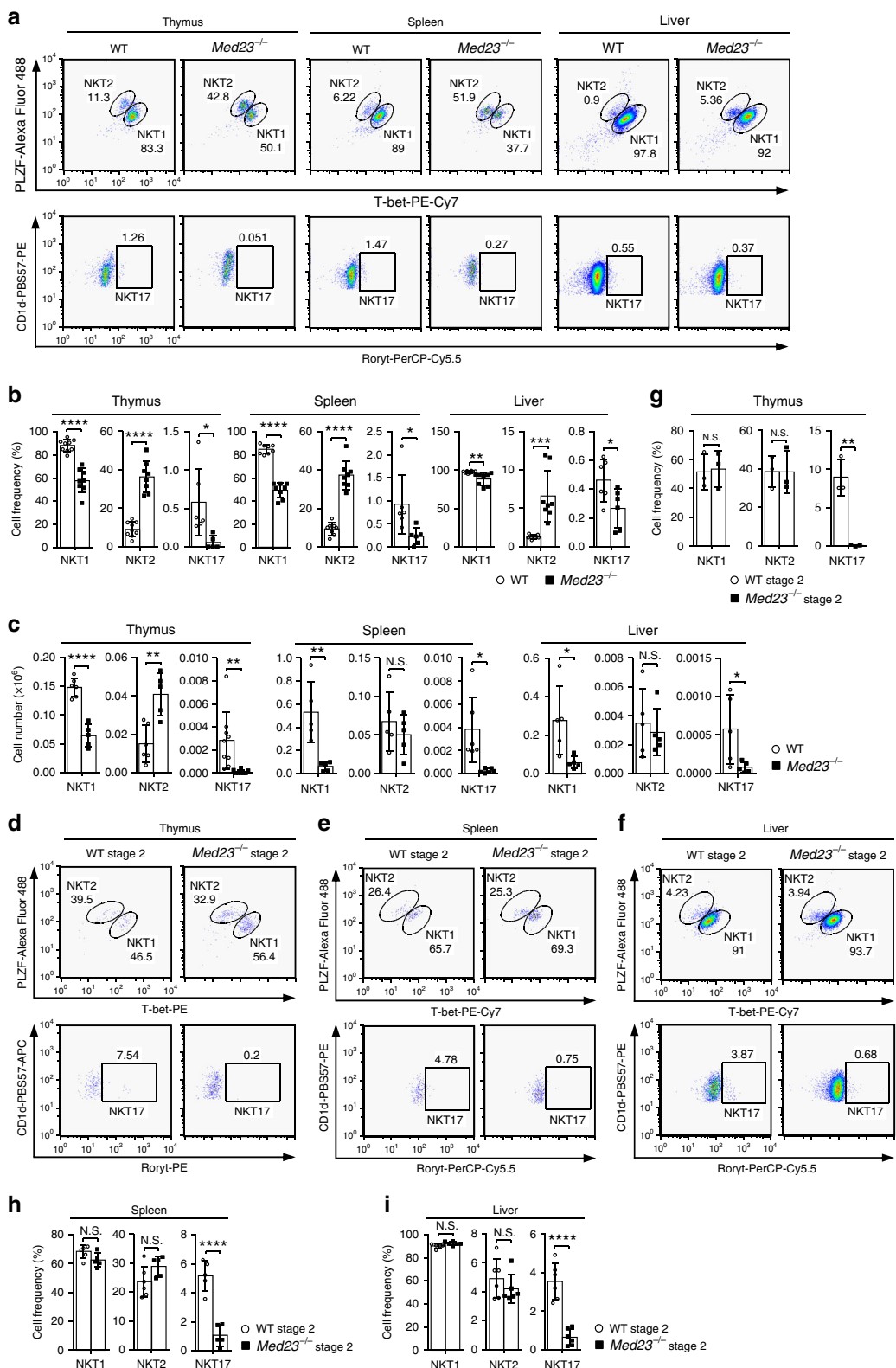

the formation of normal and cleft palates[46]. Unexpectedly, the transition from stage 2 to stage 3 upregulated all genes encoding chemokine activity (Fig. 4c), suggesting that stage 3 cells may have the ability to recruit other immune cells. Overall, stage 2 and stage 3 cells likely play different roles in the immune response. In addition to regulating the acquisition of NK cell characteristics,

the participation of Med23 in the functional transition remains to be investigated.

**Med23 facilitates function acquisition by iNKT cells**. Stage 2 and stage 3 cells exhibited a distinct cytokine bias (Fig. 4b).

**Fig. 2** *Med23*-deficient iNKT cells have impaired functional differentiation. **a** Representative dot plots of NKT1, NKT2, and NKT17 cells in the thymi, spleens, and livers of five- to eight-week-old WT and $Med23^{-/-}$ mice. **b** The percentage of NKT1, NKT2, and NKT17 cells in the thymi, spleens, and livers of WT and $Med23^{-/-}$ mice (thymic NKT1 and NKT2, WT, $n = 9$, KO, $n = 8$; thymic NKT17, WT, $n = 6$, KO, $n = 5$; splenic and liver NKT1 and NKT2, $n = 8$; splenic and liver NKT17, $n = 6$). **c** The absolute number of NKT1, NKT2, and NKT17 cells in the thymi, spleens, and livers of WT and $Med23^{-/-}$ mice (thymic NKT1 and NKT2, WT, $n = 6$, KO, $n = 5$; thymic NKT17, WT, $n = 9$, KO, $n = 8$; splenic NKT1 and NKT2, $n = 5$; splenic NKT17, $n = 6$; liver, $n = 5$). **d–f** Flow cytometric analysis of NKT1, NKT2, and NKT17 cells in stage 2 cells in the thymi (**d**), spleens (**e**) and livers (**f**) of five- to eight-week-old WT and $Med23^{-/-}$ mice. **g–i** Representative frequency of NKT1, NKT2, and NKT17 cells in stage 2 cells in the thymi (**g**), spleens (**h**) and livers (**i**) of WT and $Med23^{-/-}$ mice (thymus, $n = 3$; splenic NKT1 and NKT2, WT, $n = 6$, KO, $n = 5$; splenic NKT17, $n = 5$; liver, $n = 6$). The data are presented as the mean ± s.d. For all panels: *$P < 0.05$; **$P < 0.01$; ***$P < 0.001$; ****$P < 0.0001$ by Student's $t$-test; N.S. no significance. All data are representative of or combined from at least three independent experiments

Because WT and $Med23^{-/-}$ stage 2 iNKT cells exhibited similar global transcriptomes (Supplementary Fig. 8a, b) and Med23 did not impact the main functional subsets in stage 2 cells (Fig. 2g–i), we did not focus any further on the difference in cytokine diversity between WT and $Med23^{-/-}$ stage 2 iNKT cells. Instead, we investigated the ability of stage 2 and stage 3 cells to respond to antigens and whether Med23 regulated this process. We first intravenously injected 2 µg of α-GalCer, a specific ligand for iNKT cells, into WT and $Med23^{-/-}$ mice. The injection of α-GalCer into WT mice rapidly increased their IL-4 serum titers 4 h after injection and their IFN-γ titers 16 h after injection. $Med23^{-/-}$ mice only exhibited slight increases in these titers upon activation by the antigen compared with the WT controls (Fig. 5a), indicating an insensitive response to the antigen.

Subsequently, we examined IFN-γ and IL-4 secretion by stage 2 and stage 3 iNKT cells upon activation in vivo. Because few stage 3 cells were present in $Med23^{-/-}$ mice, we analyzed only WT stage 2 and stage 3 cells and *Med23*-deficient stage 2 cells in follow-up studies. iNKT cells in spleens and livers from WT and $Med23^{-/-}$ mice increased IFN-γ and IL-4 secretion after α-GalCer stimulation compared to the mock-treated controls (Fig. 5b, c). Splenic WT stage 3 cells produced the highest levels of IFN-γ and IL-4, whereas the cytokine secretion levels of *Med23*-deficient stage 2 cells were even lower than those of WT stage 2 cells upon activation (Fig. 5d). IFN-γ production in the liver followed the same trend (Fig. 5e). Furthermore, we found that splenic and liver WT stage 3 cells produced IFN-γ and IL-4 much faster and were more sensitive than WT stage 2 cells after α-GalCer stimulation. $Med23^{-/-}$ stage 2 cells exhibited the weakest production and antigen sensitivity with the exception of IL-4 secretion in the livers (Fig. 5f–i). Thus, the terminal maturation of iNKT cells included functional improvements that were dependent on Med23. Since TCR downmodulation of iNKT cells is gradual and recovers within 24 h after the administration of α-GalCer[47], we only detected iNKT cells that were activated for less than 2 h in vivo. Csf-2 deficiency has been reported to impair cytokine secretion in iNKT cells upon activation;[48] thus, we investigated the expression of Csf-2 in WT and $Med23^{-/-}$ stage 2 iNKT cells. High expression of Csf2 was observed in $Med23^{-/-}$ stage2 cells indicating that Med23 deletion did not impact iNKT cell function by Csf-2 (Supplementary Fig. 8c).

**Med23-deficient iNKT cells lose their recruitment capacity.** Intriguingly, the RNA-seq data showed that WT stage 3 cells expressed a diverse variety of chemokines compared with WT stage 2 cells (Fig. 4c). We investigated whether the transition from stage 2 to stage 3 improved recruitment capacity and determined the role of Med23. First, we investigated the recruitment capacity of iNKT cells in vivo. WT mice injected intravenously with 2 µg of α-GalCer exhibited a significant accumulation of neutrophils in

the liver in the first 2 h after injection, followed by a significant accumulation of B cells and NK cells, although there was a loss of cellularity among monocytes and macrophages. A slight increase in T cell numbers was also detected (Fig. 6a and Supplementary Fig. 9a, b). These results confirmed that iNKT cells were able to recruit diverse immune cells after stimulation.

One hour after stimulation with α-GalCer, we sorted the stage 2 and stage 3 cells in WT mice and stage 2 cells in $Med23^{-/-}$ mice and examined their chemokine expression levels. WT stage 3 cells had significantly higher expression of *Xcl1*, *Ccl5*, and *Ccl4* than the other cells. However, $Med23^{-/-}$ stage 2 cells were unable to maintain the same chemokine expression levels as WT stage 3 cells and exhibited significantly decreased *Ccl4* expression compared with WT stage 2 cells (Fig. 6b). We also measured the production of chemokine ligand 5 (CCL5), which regulates the recruitment of a variety of leukocytes, such as T cells and neutrophils, to sites of inflammation[49]. Splenic and liver WT iNKT cells upregulated CCL5 production after α-GalCer stimulation compared to the mock-treated controls (Fig. 6c, d). *Med23*-deficient splenic and liver stage 2 cells exhibited impaired secretion of CCL5, and WT stage 3 cells exhibited the highest increase in CCL5 production upon activation (Fig. 6e). To determine the role of CCL5 secretion by iNKT cells in cell recruitment, we injected WT mice with a CCL5 antagonist, Met-RANTES, or PBS as a control before α-GalCer administration. After activation of iNKT cells for 2 h, the cell number of neutrophils was significantly increased in the liver in PBS-treated mice. By contrast, Met-RANTES-treated mice exhibited only a modest increase compared with the PBS-treated controls (Supplementary Fig. 10). These results demonstrated that Med23 may partially impact iNKT cell recruitment capacity by regulating their CCL5 secretion. Overall, our data showed that Med23 regulates recruitment capacity, including in the terminal maturation of iNKT cells.

**Med23 ablation impairs anti-tumor iNKT cell function.** Considering that the loss of Med23 impaired the functional maturation of iNKT cells, we investigated whether Med23 disrupted their pathophysiological function. WT and $Med23^{-/-}$ mice were inoculated with B16F10 melanoma cells and intravenously injected with 2 µg of mock or α-GalCer three times (on days 0, 4, and 8), after which the number of lung metastases was counted. We observed that WT mice that received α-GalCer exhibited a marked resistance to B16F10 cell metastasis to the lungs compared with mock-injected WT mice. Although $Med23^{-/-}$ mice displayed a low capacity to prevent B16F10 cell metastasis after α-GalCer administration, the inhibitory effect was obviously decreased compared with that observed in WT mice (Fig. 7a–c). This impaired anti-tumor ability might be attributable to the decreased accumulation of iNKT cells in the lungs and damage to their production of critical anti-tumor factors (IFN-γ and CCL5)

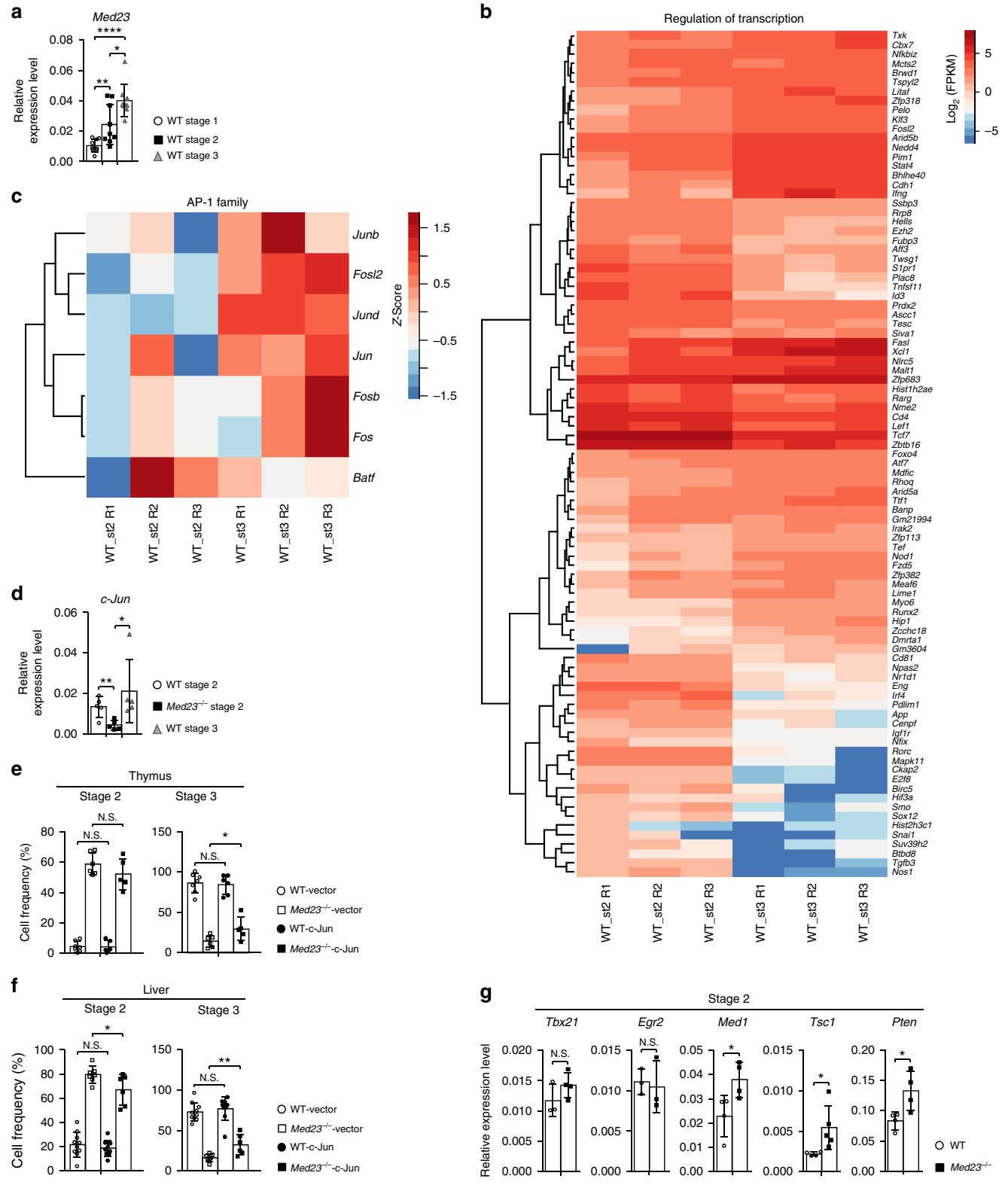

after α-GalCer administration (days 0 and 4) (Fig. 7d–g and Supplementary Fig. 1b)[50,51]. Since surface NK1.1 expression is significantly downregulated in iNKT cells 8 h after the administration of α-GalCer and decreased NK1.1 expression has previously been shown to persist for at least half a year[52], we were unable to recognize stage 2 and stage 3 iNKT cells after administering α-GalCer twice. Instead, we analyzed the frequency of IFN-γ and CCL5 secretion by total iNKT cells.

Without α-GalCer injections, $Med23^{-/-}$ mice exhibited better anti-tumor effects than did WT controls (Fig. 7a, b), probably due to the hyperactivation of T cells[26]. To exclude the influence of a low T cell activation threshold in $Med23^{-/-}$ mice, we transferred

**Fig. 3** Ectopic expression of c-Jun partially rescues the development of $Med23^{-/-}$ iNKT cells. **a** Quantitative RT-PCR analysis of $Med23$ mRNA levels in WT thymic iNKT cells at stage 1, stage 2, and stage 3 as sorted by flow cytometry ($n = 9$). **b** The expression of representative genes with known transcriptional regulation between WT stage 2 and stage 3 iNKT cells (fold change $\geq 2$ and $p$-value $\leq 0.05$). **c** The expression of genes included in the AP-1 pathway in WT thymic iNKT cells at stage 2 and stage 3. Absolute expression values were transformed into $Z$ scores before visualization. **d** c-Jun transcriptional levels in thymic iNKT cells at stage 2 and stage 3 from WT mice and stage 2 from $Med23^{-/-}$ mice ($n = 5$). **e** The percentage of iNKT cells at stage 2 and stage 3 in the thymi of active c-Jun- or vector-transduced BM chimeric mice (WT-vector, $n = 7$; KO-vector, $n = 6$; WT-c-Jun, $n = 6$; KO-c-Jun, $n = 5$). **f** The percentage of iNKT cells at stage 2 and stage 3 in the livers of active c-Jun- or vector-transduced BM chimeric mice (WT-vector, $n = 10$; KO-vector, $n = 7$; WT-c-Jun, $n = 9$; KO-c-Jun, $n = 6$). **g** Quantitative RT-PCR analysis of representative genes with known regulation in the transition from stage 2 to stage 3 ($Tbx21$, $Egr2$, $Med1$, $Tsc1$, $Pten$) in sorted thymic stage 2 iNKT cells from WT mice and $Med23^{-/-}$ mice ($Tbx21$, WT, $n = 3$, KO, $n = 4$; $Egr2$, $n = 3$; $Med1$ and $Pten$, $n = 4$; $Tsc1$, WT, $n = 4$, KO, $n = 5$). All expression levels (**a, d, g**) were normalized to $Gapdh$ expression. The data are presented as the mean ± s.d. For all panels: *$P < 0.05$; **$P < 0.01$; ****$P < 0.0001$ by Student's $t$-test; N.S. no significance. All data are representative of or combined from at least three independent experiments

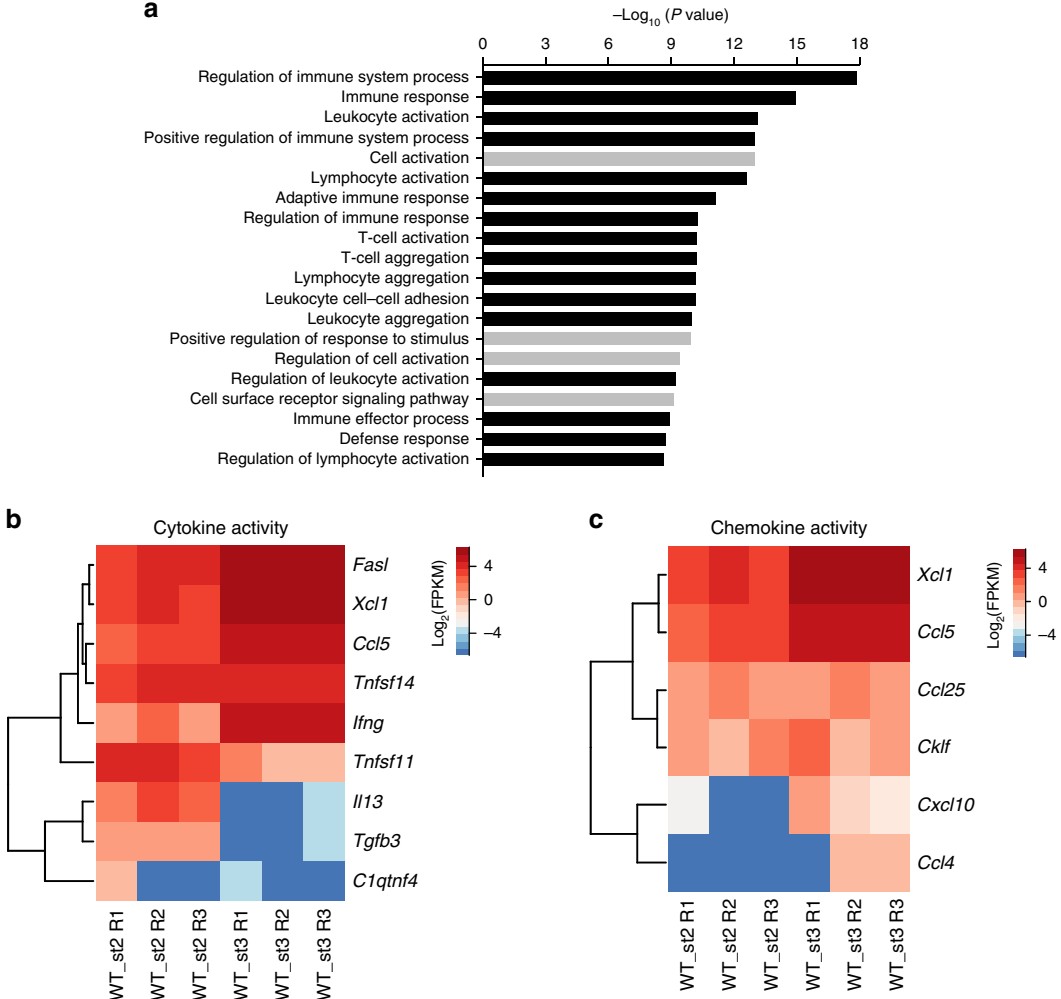

**Fig. 4** Gene expression profiles of thymic WT stage 2 and stage 3 cells. **a** Pathway enrichment analysis of differentially expressed genes (modified Fisher's exact $p$-value < 0.05) in thymic stage 2 and stage 3 iNKT cells from WT mice (black: categories related to immune process). **b** Differential expression of representative genes with known cytokine activity between WT iNKT cells at stage 2 and stage 3 (fold change $\geq 2$ and $p$-value $\leq 0.05$). **c** Heat map of chemokine activity-related gene expression in stage 2 and stage 3 iNKT cells from WT mice. All data are combined from at least three independent experiments

purified WT or $Med23^{-/-}$ iNKT cells ($2 \times 10^5$) into $J\alpha18^{-/-}$ mice (iNKT cell-deficient) that had been previously inoculated with B16F10 melanoma cells and injected three times with 2 μg of α-GalCer (days 0, 4, and 8). The mice receiving WT iNKT cells demonstrated a significantly increased ability to prevent B16F10 cell metastasis compared with the mice that received $Med23^{-/-}$ iNKT cells (Fig. 7h, i). Together, these data suggest that Med23 deficiency can impair the anti-tumor effects of iNKT cells.

## Discussion

We previously demonstrated that Med23 regulates conventional T cell activation and prevents autoimmunity[26]. Here, we found that Med23 deletion led to a specific block of iNKT cell development from stage 2 to stage 3. This phenotype provided a rare opportunity to investigate the terminal maturation of iNKT cells. The functional assay revealed that loss of Med23 impaired the acquisition of NK cell characteristics and decreased cytokine

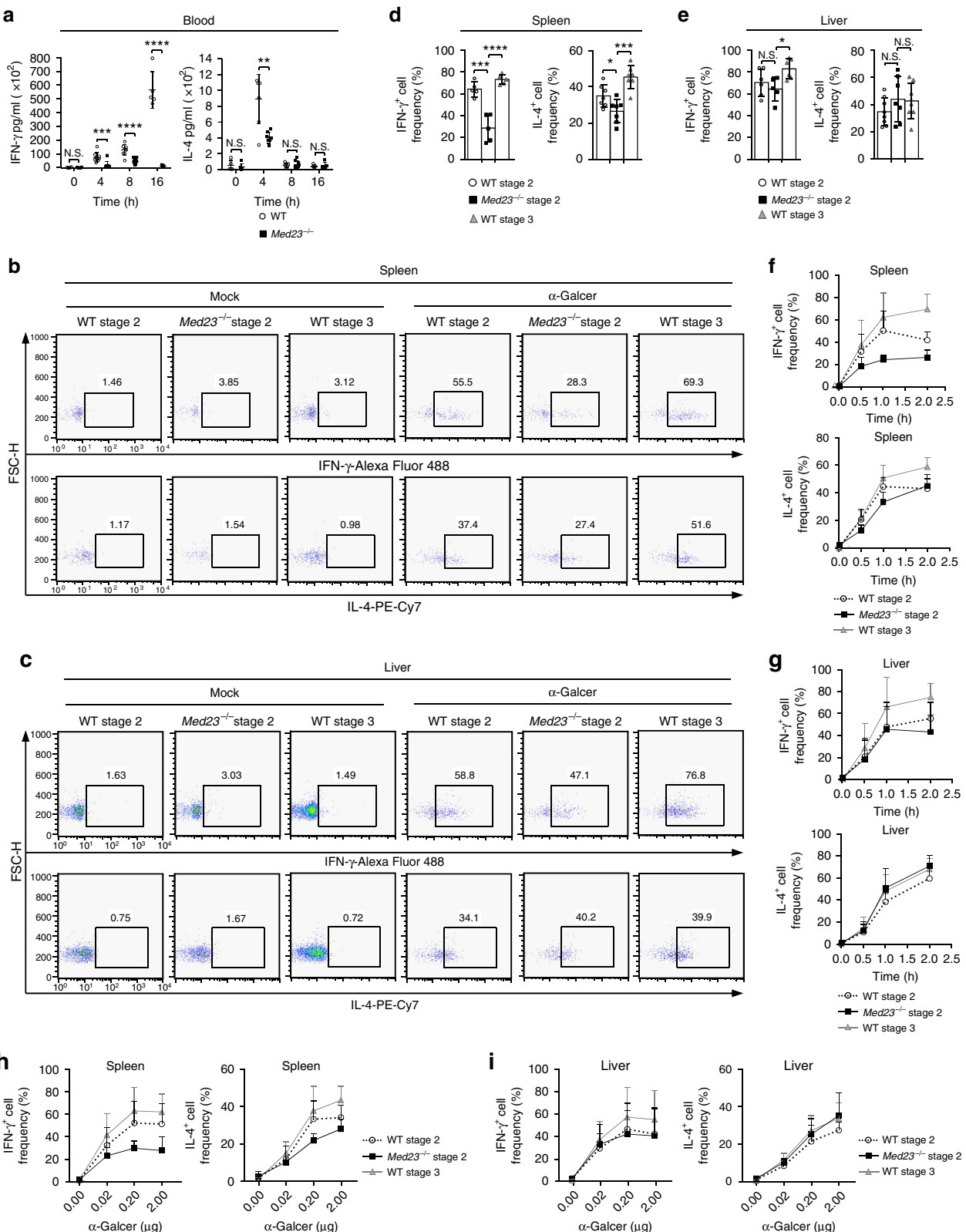

secretion and the recruitment capacity of iNKT cells. Moreover, Med23 deficiency in iNKT cells resulted in attenuated anti-tumor activity after α-GalCer administration. Collectively, our data suggest that Med23, one of the tail modules of the mediator complex that regulates specific gene transcription, functions as a

critical regulator of transcription that controls the development and functional maturation of iNKT cells.

iNKT cells finish a critical self-expansion process in the transition from stage 0 to stage 1. From stage 1 to stage 2, iNKT cells obtain a memory-like CD44[high] or activated phenotype[53].

**Fig. 5** Med23 promotes cytokine secretion by iNKT cells upon their activation. **a** ELISA of IFN-γ and IL-4 titers in the serum of WT and $Med23^{-/-}$ mice at various time points (horizontal axis) after α-GalCer administration (IFN-γ, 0 h, $n = 4$; 4 h, $n = 9$; 8 h, WT, $n = 8$, KO, $n = 9$; 16 h, $n = 5$) (IL-4, 0 h, WT, $n = 5$, KO, $n = 6$; 4 h, $n = 7$; 8 h, $n = 9$; 16 h, $n = 5$). **b, c** Production of IFN-γ and IL-4 in WT stage 2 and stage 3 cells and $Med23^{-/-}$ stage 2 cells of the spleen (**b**) and liver (**c**) after in vivo stimulation for 1 h with 2 μg of α-GalCer or mock. **d, e** The percentage of IFN-γ+ and IL-4+ cells among WT stage 2 and stage 3 cells and $Med23^{-/-}$ stage 2 cells in the spleens (**d**) and livers (**e**) 1 h after the injection of α-GalCer (spleen, IFN-γ+ cell frequency, $n = 5$; IL-4+ cell frequency, $n = 7$) (liver, IFN-γ+ cell frequency, WT Stage 2 and Stage 3, $n = 6$, KO Stage 2, $n = 5$; IL-4+ cell frequency, WT Stage 2 and Stage 3, $n = 8$, KO Stage 2, $n = 7$). **f, g** IFN-γ and IL-4 production from WT stage 2 and stage 3 cells and $Med23^{-/-}$ stage 2 cells in the spleens (**f**) and livers (**g**) at various time points (horizontal axis) after the injection of α-GalCer ($n = 5$). **h, i** The percentage of IFN-γ+ and IL-4+ cells in WT stage 2 and stage 3 cells and $Med23^{-/-}$ stage 2 cells in the spleens (**h**) and livers (**i**) after the injection of α-GalCer at various concentrations (horizontal axis) (0.00 μg, $n = 6$; 0.02 μg, 0.20 μg, and 2.00 μg, $n = 5$). The data are presented as the mean ± s.d. For all panels: *$P < 0.05$; **$P < 0.01$; ***$P < 0.001$; ****$P < 0.0001$ by Student's $t$-test; N.S. no significance. All data are representative of or combined from at least three independent experiments

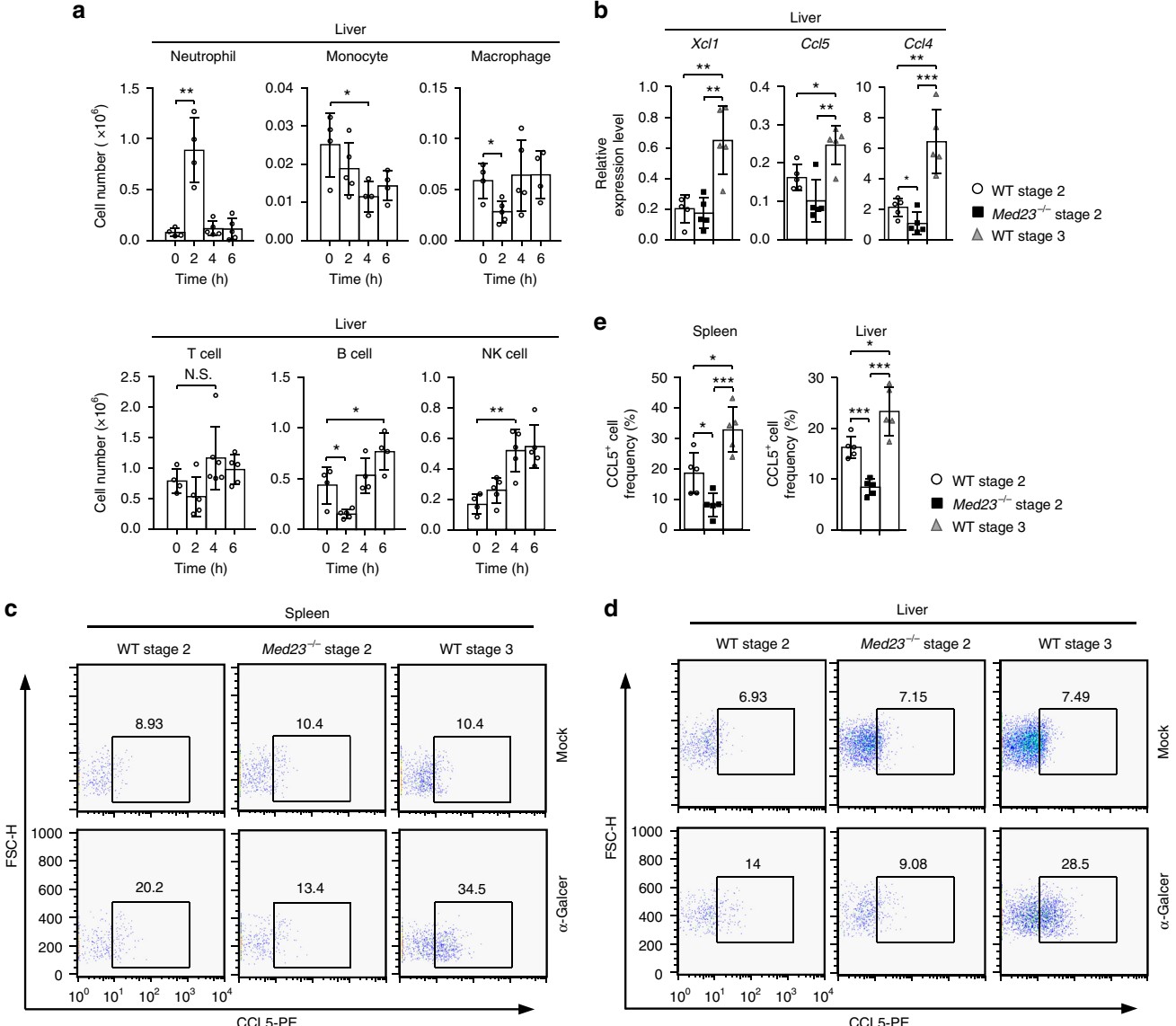

**Fig. 6** The recruitment capacity of iNKT cells is regulated by Med23. **a** The absolute numbers of neutrophils, monocytes, macrophages, T cells, B cells and NK cells in the livers of eight-week-old WT mice injected with 2 μg of α-GalCer at various time points (neutrophil, 0 h and 2 h, $n = 4$; 4 h and 6 h, $n = 5$) (monocyte, 0 h, 4 h and 6 h, $n = 4$; 2 h, $n = 5$) (macrophage, 0 h and 6 h, $n = 4$; 2 h and 4 h, $n = 5$) (T cell, 0 h, $n = 4$; 2 h and 6 h, $n = 5$; 4 h, $n = 6$) (B cell, 0 h, 4 h and 6 h, $n = 4$; 2 h, $n = 5$) (NK cell, 0 h, $n = 4$; 2 h, 4 h and 6 h, $n = 5$). **b** The transcriptional levels of $Xcl1$, $Ccl5$, and $Ccl4$ in sorted WT stage 2 and stage 3 cells and $Med23^{-/-}$ stage 2 cells from the liver 1 h after injection of 2 μg of α-GalCer. Gene expression was normalized to $Gapdh$ expression ($n = 5$). **c, d** Production of CCL5 in WT stage 2 and stage 3 cells and $Med23^{-/-}$ stage 2 cells of the spleen (**c**) and liver (**d**) 1 h after the injection of 2 μg of α-GalCer or mock. **e** The percentage of CCL5+ cells among WT stage 2 and stage 3 cells and $Med23^{-/-}$ stage 2 cells in the spleens and livers 1 h after the injection of 2 μg of α-GalCer ($n = 5$). The data are presented as the mean ± s.d. For all panels: *$P < 0.05$; **$P < 0.01$; ***$P < 0.001$ by Student's $t$-test; N.S. no significance. All data are representative of or combined from at least three independent experiments

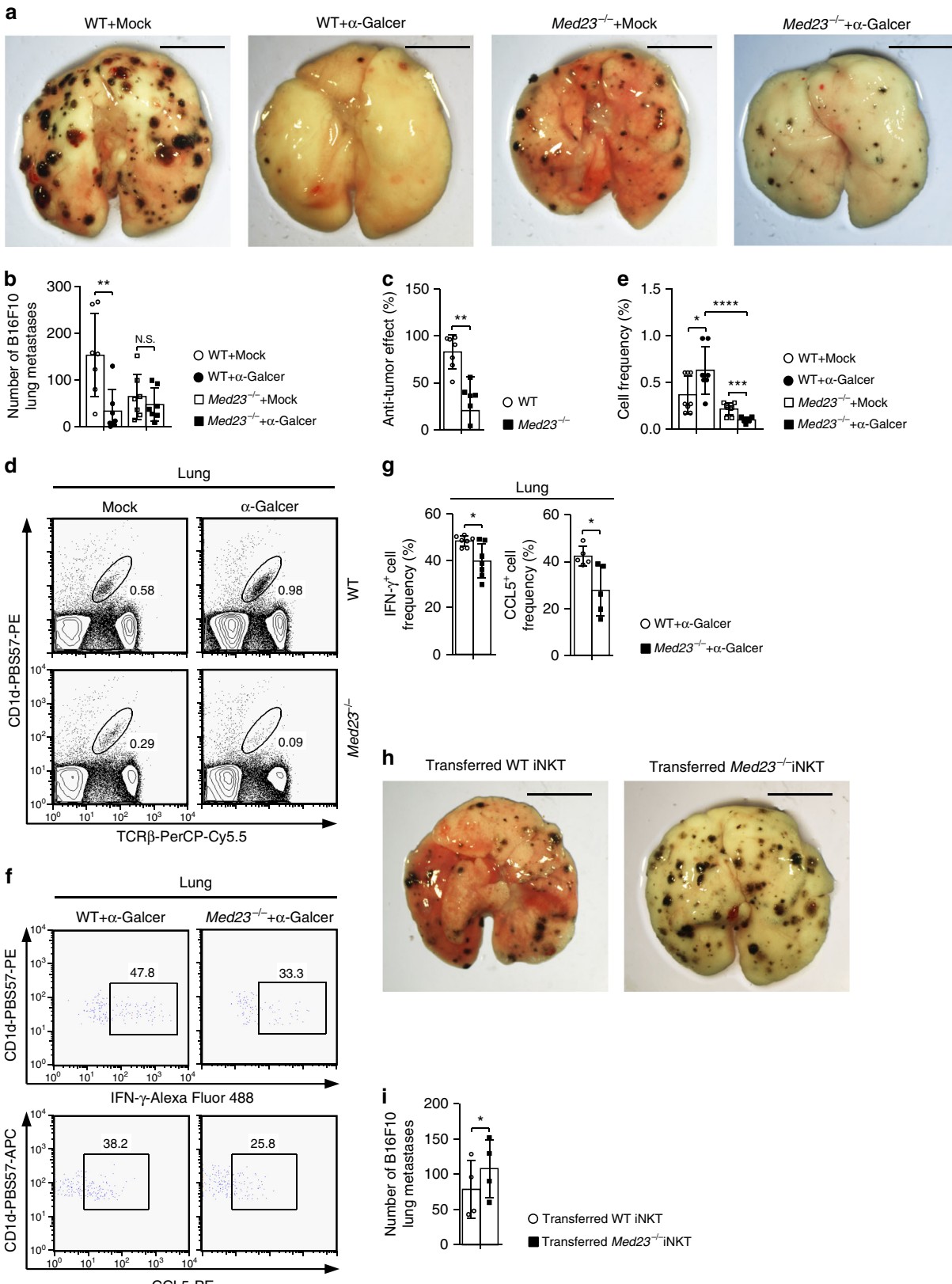

However, the process that occurs during the transition from stage 2 to stage 3 is largely unknown, with the exception of the expression of certain NK cell-related markers. Our study summarizes the characteristics of stage 2 and stage 3 cells in WT mice using global transcriptional analysis and functional assays. Stage 3 cells upregulated many NK cell-related markers, produced cytokines faster and were more sensitive than stage 2 cells after stimulation with antigens. Moreover, stage 3 cells gained

**Fig. 7** Med23 deficiency impairs the anti-tumor function of iNKT cells. **a–c** WT and $Med23^{-/-}$ mice were inoculated with $2 \times 10^5$ B16F10 cells by i.v. injection. On the same day and again on days 4 and 8, WT and $Med23^{-/-}$ mice received 2 μg of α-GalCer or mock by i.v. injection. Fourteen days after tumor inoculation, the lungs were harvested (**a**), B16F10 colonies were counted (**b**) and the anti-tumor effects of α-GalCer were assessed (**c**) ($n = 7$). Scale bar: 0.5 cm. The anti-tumor effect (%) = 1-B16F10 colonies (α-GalCer administration)/B16F10 colonies (mock administration). **d, e** Flow cytometry data (**d**) and the percentage (**e**) of iNKT cells in the lungs of WT and $Med23^{-/-}$ mice with post-B16F10 inoculation and α-GalCer or mock administration on days 0 and 4 (WT + Mock, KO + Mock and KO + α-GalCer, $n = 8$; WT + α-GalCer, $n = 7$). **f, g** After in vitro stimulation with PMA and ionomycin for 2 h, flow cytometric analysis (**f**) was performed, and the percentage (**g**) of IFN-γ$^+$ and CCL5$^+$ cells among iNKT cells in the lungs of WT and $Med23^{-/-}$ mice with post-B16F10 inoculation and α-GalCer administration on days 0 and 4 was determined (IFN-γ$^+$ cell frequency, $n = 7$; CCL5$^+$ cell frequency, $n = 5$). **h, i** $J\alpha18^{-/-}$ mice were inoculated with $2 \times 10^5$ B16F10 cells by i.v. injection. On the same day and again on days 4 and 8, the $J\alpha18^{-/-}$ mice received $2 \times 10^5$ liver-derived iNKT cells from WT or $Med23^{-/-}$ mice by i.v. injection with 2 μg of α-GalCer. Fourteen days after tumor inoculation, the lungs were harvested (**h**), and B16F10 colonies were counted (**i**) ($n = 4$, paired $t$-test). Scale bar: 0.5 cm. The data are presented as the mean ± s.d. For all panels: *$P < 0.05$; **$P < 0.01$; ***$P < 0.001$; ****$P < 0.0001$ by Student's $t$-test; N.S. no significance. All data are representative of or combined from at least three independent experiments

recruitment capacity by increasing the expression of a diverse number of chemokines. Collectively, our study provides a better understanding of terminal maturation in iNKT cells, highlighting the improved function of stage 3 cells.

T cell development and function are largely dependent on transcription factors[54,55]. Different transcription factors regulate unique physiological processes. The main role of Med23, an important component of the mediator complex, is to regulate gene transcription. Med23 is required specifically for mediator recruitment to a subset of immediate early response genes that are predominantly controlled by the mitogen-activated protein kinase (MAPK) signaling pathway, such as the Egr family of proteins, c-Fos, and c-Jun[21,22]. In our study, the transition from stage 2 to stage 3 reprogrammed the transcriptional regulation profile, highlighting the significance of transcription in this process. Unexpectedly, the expression of many transcription factors related to Med23 were altered during this transition. The expression of AP-1 transcription factors differed between WT stage 2 and stage 3 iNKT cells (Fig. 3c). The roles of AP-1 activity in iNKT cell development have been studied in three genetically manipulated mouse models: BATF-transgenic, Fosl2, and JunB knockout mice. The loss of Fosl2 in iNKT cells significantly increases c-Jun expression and the cellularity and cytokine secretion bias of iNKT cells[56–58]. Correspondingly, the expression of c-Jun, a key component of the AP-1 complex, was significantly downregulated in $Med23$-deficient stage 2 iNKT cells, and the ectopic expression of c-Jun in $Med23^{-/-}$ iNKT cells partially restored the stage 2 to stage 3 transition (Fig. 3d–f). Based on these and other published data, Med23 may regulate iNKT cell development from stage 2 to stage 3 by impacting the transcription of specific genes. Whether Med23 controls this final transition through other potential regulators, such as RUNX2 and the FOXO family, remains to be elucidated.

It has been reported that the transition from stage 2 to stage 3 is regulated by many other regulators. Med1 is another subunit of the mediator complex; its deficiency impairs the development of iNKT cells from stage 0 and reduces IL-2Rβ and T-bet expression[34]. $Hobit^{-/-}$ iNKT cells, which have a low percentage of the mature population, enhance IFN-γ production after α-GalCer administration[59]. Due to loss of the vitamin D receptor, iNKT cells fail to increase NK1.1 and Ly49C expression but have normal levels of CD122 and NKG2A/C/E expression[60]. IL-15 plays an important role in iNKT cell development and survival[30]. In our study, Med23 deficiency distinct blocked iNKT cell development at stage 2 (Fig. 1d–f) and further impaired the recruitment capacity of iNKT cells (Fig. 6b–e). Taken together, Med23 displays dominance and specificity in the regulation of iNKT cell terminal maturation compared to other regulators.

Decreased numbers and impaired function of iNKT cells are closely connected to poor clinical outcomes in patients with malignant diseases[61,62]. Recently, more researchers have focused on the anti-tumor activity of iNKT cells. iNKT cells are stimulated by α-GalCer, a specific agonist for mouse and human iNKT cells[7,63]. The stimulated iNKT cells then induce cytokine production and subsequently activate other immune cells, such as NK cells, CD8$^+$ CTLs, and dendritic cells, against various tumors, such as melanoma, colon, and hematopoietic cancers[8,64,65]. The decreased secretion of IFN-γ and CCL5 in $Med23$-deficient iNKT cells after α-GalCer administration may be one of the main reasons for their impaired anti-tumor ability. These results imply a critical role for stage 3 iNKT cells in tumor therapy and emphasize that Med23 controls the final maturation of iNKT cells, which is closely related to their anti-tumor effects.

Our results support a new paradigm regarding the terminal maturation of iNKT cells. This process not only includes functional subset differentiation and abundant receptor expression correlated with NK cells but also improvement in cytokine secretion and recruitment capacity upon activation. This transition is intimately associated with the role of iNKT cells in disease. In contrast to NK cells, mature iNKT cells possess a distinct antigen that is activated in addition to producing copious levels of several cytokines after stimulation and expressing NK cell markers. Additionally, mature iNKT cells are not the same as conventional T cells, are activated without priming and possess recruitment capacity. These features suggest that Med23 is a key regulator that controls the terminal maturation of iNKT cells and determines their final destiny.

## Methods

**Mice**. $Med23^{fl/fl}$ mice had been obtained from Professor G. Wang (Shanghai Institute of Biochemistry and Cell Biology, Chinese Academy of Sciences) and backcrossed to a B6 background for at least 10 generations unless otherwise indicated[26]. $Cd4$-$Cre$ transgenic mice were obtained from Professor Z. Hua (Nanjing University). $Vav1$-$Cre$ transgenic mice (strain: B6.Cg-$Commd10^{Tg (Vav1-icre) A2Kio}$/J) and CD45.1 mice (strain: B6.SJL-$Ptprc^aPepc^b$/BoyJ) were purchased from The Jackson Laboratory. $J\alpha18^{-/-}$ mice were originally obtained from Professor L. Bai (University of Science and Technology of China). $Rag2^{-/-}$ mice were obtained from the Institute of Development Biology and Molecular Medicine (Fudan University) and maintained on a B6 background. Both male and female mice between 5 and 12 weeks of age were used for all experiments. All mice were maintained under pathogen-free conditions and genotyped by PCR before experimentation. Mice were randomly allocated to experimental groups and processed to ensure the robustness of our conclusions. We were not blinded to the group allocation during experiments and outcome assessment. All animal experiments were conducted according to the National Institutes of Health guidelines and were approved by the Institutional Animal Care and Use Committee of the Shanghai Institute of Biochemistry and Cell Biology, Chinese Academy of Sciences. The sequences of primers used for genotyping are listed in the Supplementary Table 1.

**Cell culture**. The PlatE cell line was obtained from Professor G. Pei (Shanghai Institute of Biochemistry and Cell Biology, Chinese Academy of Sciences) and was originally purchased from Cell Biolabs, Inc. The B16F10 melanoma cell line was obtained from the Cell Bank (Shanghai Institute of Biochemistry and Cell Biology, Chinese Academy of Sciences). The cells were cultured in a humidified incubator at

37 °C, 5% $CO_2$ in DMEM (Gibco) supplemented with 10% FBS (Gibco) and 1% penicillin/streptomycin (Gibco). All cell lines used were tested and shown to be mycoplasma-free.

**Antibodies and reagents**. Fluorescently conjugated protein or antibodies used for cell-surface staining and intracellular staining for cytokines and transcription factors are listed in Supplementary Table 2. α-GalCer (KRN7000) was purchased from Enzo. CD1d-PBS57 (conjugated with either PE or allophycocyanin) was obtained from the tetramer facility of the US National Institutes of Health. Collagenase IV used for tissue digestion was purchased from Sigma. Taq Master Mix was purchased from Vazyme Biotech. A mouse IFN-γ ELISA kit and a mouse IL-4 ELISA kit were obtained from BioLegend. PMA and ionomycin were obtained from Merck. Stem cell factor, murine IL-3, and murine IL-6 were obtained from PeproTech. Met-RANTES was purchased from R&D Systems. DAPI was obtained from Cell Signaling Technology.

**Tissue preparation and cell isolation**. Single cell thymocyte suspensions were obtained by triturating and filtering through a nylon screen. Splenocytes were prepared by squeezing, followed by red blood cell lysis before filtration. The lungs or livers were perfused with cold PBS before extraction to remove the blood cells. Then, the lungs were cut into pieces and incubated with shaking (200 rpm) at 37 °C for 90 min in RPMI medium (Gibco) containing 5% FBS (Gibco), 160 μg ml$^{-1}$ collagenase IV (Sigma) and 0.2 μg ml$^{-1}$ DNase I (Shanghai Sanjie). The livers were minced and filtered through a cell strainer (40 μM; BD Biosciences). Leukocytes were isolated from the digested tissues or liver cell suspensions by density fractionation using discontinuous 40–70% (vol/vol) Percoll (GE Healthcare) gradients. Bone marrow cells were obtained by flushing the tibias and femurs and then lysing the red blood cells before filtration.

**FACS analysis and cell sorting**. For cell surface staining, the cells were distributed in 5 ml polystyrene round-bottom tubes (BD Biosciences) and stained for 40 min at 4 °C with the indicated antibodies[66]. Intracellular staining for cytokines was performed after 10 min of fixation with 2% formaldehyde solution in PBS at room temperature and 5 min of permeabilization in Perm/Wash Buffer (BD Biosciences) at 4 °C. Intracellular staining for transcription factors (PLZF, T-bet and RORγt) was performed using a Foxp3 staining kit (eBioscience) according to the manufacturer's protocol. Stage 2 or stage 3 iNKT cells from the thymus or liver and total iNKT cells from the liver were sorted by FACS for RNA-seq analysis, relative gene transcription analysis, or adoptive transfers. To obtain stage 2 or stage 3 iNKT cells from the thymus for relative gene transcription analysis, cell suspensions were depleted of CD8$^+$ (53–6.7) cells before sorting using biotinylated antibodies (BioLegend, 100704) bound to magnetic streptavidin beads (Life Technologies). Cell fluorescence was performed on a four-laser BD LSRFortessa II or a two-laser BD FACSCalibur, and the acquired data were analyzed with FlowJo software (TreeStar, Inc., Olten, Switzerland). Cell sorting was performed with a BD FACSAria II after surface staining. The sorted cell purity was greater than 95%.

**RNA-seq, library generation and bioinformatics analysis**. RNA was extracted, purified and checked for integrity using an Agilent Bioanalyzer 2100 (Agilent Technologies, Santa Clara, CA, US). Libraries were generated for sequencing using a SMARTer Stranded Total RNA-Seq Kit—Pico Input Mammalian (Illumina). Libraries were sequenced using an Illumina HiSeq X Ten sequencer. After quality filtration and the removal of sequencing adapters with Trimmomatic (version 0.36), the RNA-seq reads were aligned to the mouse genome (mm10) using HISAT2[67], and only reads hitting no more than 2 genomic loci were used to quantify gene expression with StringTie[68]. A gtf file of mouse genes was downloaded from GENCODE (vM12). Gene expression was normalized to FPKM. The fold change and p-value of all detected genes were estimated by Ballgown[69]. Only genes with a fold change ≥ 2 and p-value ≤ 0.05 were considered differentially expressed. Gene Ontology biological processes that were enriched based on the differentially regulated genes were identified by DAVID (modified Fisher exact p-value < 0.05)[70]. All data are representative of three independent experiments.

**In vivo responses after α-GalCer administration**. Mice were injected intravenously with 2 μg of α-GalCer or a mock. At different time points, blood samples were collected from the postcava, and cytokines were detected in the serum. Alternatively, mice were sacrificed, spleens and livers were harvested, single cell suspensions were prepared and the cells were cultured for 2 h with brefeldin A (eBioscience) to inhibit cytokine secretion and then stained with fluorescently-labeled antibodies and analyzed by FACS.

**Met-RANTES treatment**. WT mice were treated with 25 μg of Met-RANTES diluted in PBS by i.p. injection. Alternatively, control mice received a similar volume of sterile PBS.

**Quantitative real-time PCR analysis**. Total RNA was extracted with TRIzol (Invitrogen) and reverse-transcribed using the SuperScript III First-Strand

Synthesis System (Invitrogen). The mRNA levels of the indicated genes were normalized to GAPDH using real-time PCR (Rotor Gene 6000; Corbett Life Sciences). SYBR Green QPCR Master Mix (Toyobo) was used for gene amplification and detection. The sequences of qPCR primers are listed in the Supplementary Table 1.

**Generation of mixed bone marrow chimeras**. Recipient $Rag2^{-/-}$ mice were subjected to semi-lethal doses of irradiation (6.0 Gy) and received a 1:1 mixture of bone marrow cells (total: $4 \times 10^6$ cells) from WT (CD45.1$^+$) and $Med23^{-/-}$ (CD45.2$^+$) mice 8 h later. The mixed bone marrow chimeras were analyzed 8 weeks after transplantation.

**Retroviral production and transduction**. Retrovirus was packaged by transfecting the PlatE cell line with pMCs-c-Jun-IRES-GFP and culturing the transfected cells in DMEM (Gibco) containing 10% FBS. The viral supernatants were collected 4 days after transfection. For retroviral transductions, whole bone marrow cells treated with 5-FU (Sigma) were seeded for 24 h in 48-well plates (Corning, Inc.) with IMDM (Gibco) containing 10% FBS, 50 μM 2-mercaptoethanol, 100 IU ml$^{-1}$ penicillin, 100 μg ml$^{-1}$ streptomycin (Gibco), 50 ng ml$^{-1}$ stem cell factor, 10 ng ml$^{-1}$ murine IL-3, 10 ng ml$^{-1}$ murine IL-6 and suspended with retroviral supernatant and 8 μg ml$^{-1}$ polybrene (Sigma). Then, the cells were centrifuged at 1500g for 2 h at 32 °C. After the second transfection, the bone marrow cells were injected intravenously into irradiated (8.0 Gy) C57BL/6 mice, and the development of iNKT cells was analyzed 8 weeks later.

**B16F10 lung metastasis model**. WT and $Med23^{-/-}$ mice received $2 \times 10^5$ B16F10 cells by i.v. injection. On the same day and on days 4 and 8, WT and $Med23^{-/-}$ mice were injected with 2 μg of α-GalCer or the mock. On day 14 after inoculation, surface lung metastases were counted. Alternatively, on day 8, WT and $Med23^{-/-}$ mice were sacrificed, and their lungs were harvested. After isolating the leukocytes from the lungs, the cells were cultured with PMA (50 ng ml$^{-1}$), ionomycin (1 μg ml$^{-1}$) and brefeldin A (1000×) for 2 h before they were stained intracellularly for cytokines.

$Jα18^{-/-}$ mice were inoculated with $2 \times 10^5$ B16F10 cells by i.v. injection. After 6 h, the mice received $2 \times 10^5$ liver-derived iNKT cells from WT or $Med23^{-/-}$ mice by i.v. injection accompanied by 2 μg of α-GalCer by i.p. injection on the same day and on days 4 and 8. B16F10 colonies were counted 14 days after tumor inoculation.

**Statistical analyses**. Statistical analyses were performed with GraphPad Prism6. All experiments were performed at least three times. Data are expressed as the mean ± s.d. and a two-tailed unpaired Student's t-test was used, unless otherwise indicated, to determine statistical significance. For all experiments: *$P < 0.05$; **$P < 0.001$; ***$P < 0.0001$, ****$P < 0.0001$.

## Data availability

The authors declare that the data supporting the findings of this study are available within the article and its supplementary information files or are available from the corresponding author upon request. All sequencing data that support the findings of this study have been deposited in the Gene Expression Omnibus (GEO) of the National Center for Biotechnology Information (NCBI) under accession number GSE107642.

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

## Acknowledgements

We would like to thank Professor G. Wang for *Med23*<sup>fl/fl</sup> mice support, Baojin Wu for animal husbandry support, Wei Bian for cell-sorting support, the tetramer facility at the US National Institutes of Health for CD1d-PBS57 support, and the National Center for Protein Science Shanghai for animal irradiation. This work was financially supported by the National Natural Science Foundation of China (Grant nos. 31530021, 31621003, 91542122, and 31500717), the Strategic Priority Research Program of the Chinese Academy of Sciences (Grant no. XDB19000000), the Youth Innovation Promotion Association of Chinese Academy of Sciences and the China Postdoctoral Science Foundation (Grant no. 2015M581672).

## Author contributions

Y.X. performed most of the work and has the right to be listed first in bibliographic documents. Y.X. and X.L. designed, performed, and analyzed all experiments. Y.S. designed, performed, and analyzed the experiments. Y.D. performed experiments. H.S and H.L. helped with mouse construction and breeding. R.L. and H.Z. generated the library and analyzed RNA-seq data under the supervision of L.W. X.Z. analyzed the experiments. Y.X. and X.L. prepared the manuscript. X.L. conceptualized the research and directed the study.

## Additional information

**Competing interests:** The authors declare no competing interests.

