## [Peer Review File · Nature Communications]

Reviewers' comments:

Reviewer #1 (Remarks to the Author):

NKT cells differentiate into five distinct subsets to perform a myriad of immune functions that mirror T helper cell functions. The subset-specific markers and transcription factors are quite well defined but the gene regulatory network (GRN) downstream of the subset-specific transcription factors are poorly, if all understood. It is in this regard that the current report significantly advances our understanding of NKT cell development and function. In this manuscript, Liu and co-workers report that Med23, a component of the Mediator complex, plays a critical role in effector differentiation of the semi-invariant natural killer T (NKT cells) as it transits from stage 2 to stage 3 in development. They demonstrate that conditional deletion of Med23 during late CD4+8+ thymocyte stage prevents the formation of stage 3 cells, which essentially are NKT1 cell subset. Curiously, however, thymic, splenic and hepatic stage 2 NKT cells express Tbx21 (Tbet), the NKT1-specific transcription factor, and produce intracellular interferon-gamma (IFN γ), albeit at lower levels when compared to Med23 sufficient stage 2 and stage 3 NKT cells. Even though they express Tbx21, they do not express inhibitory and stimulatory receptors characteristic of NK cells. Although Med23-deficient stage 2 NKT cells produce intracellular IFN γ , they do not secrete it. Additionally, T cell-specific Med23 deficiency upregulates Zbtb7b (Th-POK) and, hence, prevents the differentiation of the NKT17 cell subset. These deficiencies in Med23-deficient NKT cell development and function impair immune surveillance against B16F10 melanoma metastasis.

The work presented is well thought out and carefully executed. The emerging data are carefully described. Nonetheless, there are a few minor concerns, which if addressed, will enhance the work substantially.

1. The inability to transit stage 2 to stage 3 of NKT cell development is reminiscent of NKT cell development in IL-15 deficient mice; see reports by Kennedy et al. *JEM*, Gapin and Kronenberg *Nat Immunol*; and Gordy et al. *J Immunol*. A comment or two in this regard could give context to this work in relation to the literature.
2. The inability to secrete cytokines despite activation-induced intracellular cytokine protein production is reminiscent of Csf-2 deficient NKT cells: see Bezbradica et al. *Immunity* 2005. A commentary connecting this work to the current report could be useful.
3. Because vav-cre mediated Med23 deletion in haematopoietic cells did not alter Klrb1c (NK1.1) expression, it was concluded that Klrb1c expression is not directly controlled by Med23. Nonetheless, it appears as though Med23 deficient NKT cells do undergo terminal effector maturation but do not express the stimulatory and inhibitory NK receptors. So, the defect may be not so much stage 2 to stage 3 maturation but one of an inability to express stage 3/NKT1-specific surface markers. Over and above this defect lies the inability to secrete stage-specific effector cytokines and the deficiency in NKT17 subset.
4. The transcriptome of stage 2/NKT2, NKT17 and stage 3/NKT1 cells are well defined independently by the groups Brenner and Kronenberg (cite the latter that appeared in *nat immunol*). Therefore, it appears unusual to have redone the transcriptome of wild type stage 2 and stage 3 NKT cells without including Med23-deficient stage 2 cells thereby forfeiting the opportunity to define a Med23-controlled GRN that promotes stage 2 to stage 3 transition.
5. Is Med23 expression controlled by Tbet?
6. The manuscript dwells on the functional deficiency caused by the afore stated defects but overlooked the opportunity to dissect the functional capabilities of NKT2 cells.
7. Reviews 1 and 11 are over a decade old; see Kumar et al. *Front Immunol* 2017 for a more

recent review.

8. Include Egawa et alii Immunity 2005, and Gapin et alii Nat Immunol 2002, along with the current ref. 10.

9. The manuscript is reasonably written considering the fact that it is written by non-native Authors. Nonetheless, there are several unusual usages of words and phrases. As well, there are grammatical errors, mostly of multiple tenses within the same sentence.

Reviewer #2 (Remarks to the Author):

Although the work has been done in a thorough manner, a main issue with this report is the limited novelty of the findings. There is already a report on a defect in NKT cells from the knockout of a different subunit (Med1) of the mediator complex. (Yue et al PNAS 2011). These authors showed that the deficient mice exhibited defects in T-bet and CD122 expression, which would correlate with a decrease in stage 3 or NKT1 cells.

A second problem is the use of markers such as NK1.1 and CD44 to separate stage 1,2, 3 cells, designating maturation steps. This is acceptable in the thymus, although the consensus is that stage 2 NKT thymocytes are complex, because they contain precursors as well as real NKT2 cells. It has not been possible to reliably separate the precursors from NKT2 cells. In the periphery, however, it is uncertain if stage 1 and stage 2 immature NKT cells are exported from the thymus. Therefore, peripheral "stage 2" NKT cells likely are mostly NKT2 cells or an impure mixture rather than a true subset. Consistent with this, some of the stage 2 cells express high amounts of T-bet (Fig. 2) and therefore may not be immature at all.

Detailed comments:

1. The authors should compare their RNA-Seq results to previous ones in which NKT1, NKT2 and NKT17 cells were analyzed. A PCA analysis of the data here combined with published data might be useful in showing how closely stage 2 NKT cells are to NKT2 cells that were selected using different surface markers.

2. Differences in the experimental protocols in Figs. 5b and 5d need to be more clearly explained in the text and figure legends, considering that the percent of NKT cells producing cytokine is different in the two panels.

3. It has been shown that localization of NKT cell subsets within the tissues influences their cytokine responses (YJ Lee et al, Immunity 2015). For example, intravenous injection of α GalCer activated NKT1 cells in spleen (and liver) which locate in red pulp whereas NKT2 cells in the T cell zone were less responsive. The data in Fig. 5 are consistent with this idea, meaning a higher proportion of NKT2 cells, rather than a general defect in the ability of the NKT cells to be activated by antigen. Do the Med23 deficient NKT cells have a defect in TCR signaling as well? Figure 5f-i suggest this could be the case, although the differences are not great and NKT2 cells were not adequately purified. What is their response to PMA+ionomycin or IL12+IL18 stimulation?

4. The authors should note that the data in Fig. 6 do not establish a causative link between cell recruitment and CCL5 expression by NKT cells. CCL5KO or knockdown NKT cells would be needed to assess this in a transfer system.

5. Some of the data have high error bars such as thymus spleen cell numbers in Fig. 1 b or thymus cells number in figure 2f. A scatter gram showing the values from individual mice would cause some clutter, but would be more revealing than a bar graph. Was the appropriate statistical test done?

6. On page 9 a reference to fig. 4d-I should read: fig. 2d-i.

Reviewers' comments:

Reviewer #1 (Remarks to the Author):

NKT cells differentiate into five distinct subsets to perform a myriad of immune functions that mirror T helper cell functions. The subset-specific markers and transcription factors are quite well defined but the gene regulatory network (GRN) downstream of the subset-specific transcription factors are poorly, if all understood. It is in this regard that the current report significantly advances our understanding of NKT cell development and function. In this manuscript, Liu and co-workers report that Med23, a component of the Mediator complex, plays a critical role in effector differentiation of the semi-invariant natural killer T (NKT cells) as it transits from stage 2 to stage 3 in development. They demonstrate that conditional deletion of Med23 during late CD4+8+ thymocyte stage prevents the formation of stage 3 cells, which essentially are NKT1 cell subset. Curiously, however, thymic, splenic and hepatic stage 2 NKT cells express Tbx21 (Tbet), the NKT1-specific transcription factor, and produce intracellular interferon-gamma (IFN γ), albeit at lower levels when compared to Med23 sufficient stage 2 and stage 3 NKT cells. even though they express Tbx21, they do not express inhibitory and stimulatory receptors characteristic of NK cells. Although Med23-deficient stage 2 NKT cells produce intracellular IFN γ , they do not secrete it. Additionally, T cell-specific Med23 deficiency upregulates Zbtb7b (Th-POK) and, hence, prevents the differentiation of the NKT17 cell subset. These deficiencies in Med23-deficient NKT cell development and function impairs immune surveillance against B16F10 melanoma metastasis.

The work presented is well thought out and carefully executed. The emerging data are carefully described. Nonetheless, there are a few minor concerns, which if addressed, will enhance the work substantially.

We thank the reviewer for his/her positive comments and valuable suggestions on our study. Improvements have been made accordingly in the revised manuscript.

1. The inability to transit stage 2 to stage 3 of NKT cell development is reminiscent of NKT cell development in IL-15 deficient mice; see reports by Kennedy et al. JEM, Gapin and Kronenberg Nat Immunol; and Gordy et alii J Immunol. A comment or two in this regard could give context to this work in relation to the literature.

We thank the reviewer's suggestion. Because only few stage 3 iNKT cells were present in *Med23*^{-/-} mice, we cannot investigate the influence of IL-15 stimulation on WT and *Med23*^{-/-} stage 3 iNKT cells directly. Alternatively, we measured the expression of specific IL-15 receptor IL-15R α in *Med23*-deficient stage 2 iNKT cells. As shown in Supplementary Figure 4a, no significant difference in IL-15R α expression has been found between WT and *Med23*-deficient stage 2 iNKT cells.

Since IL-15 is important for survival of iNKT cells^{1,2}, we also investigated the levels of apoptosis in WT and *Med23*-deficient stage 2 iNKT cells. As seen in Supplementary Figure 4b and c, loss of *Med23* did not impact the frequency of Annexin V⁺DAPI⁻ stage 2 iNKT cells. Thus, our data indicate that *Med23* does not regulate iNKT cell development by IL-15. Notably, the cell number of Annexin V⁺DAPI⁻ stage 2 iNKT cells were increased due to the high percentage of stage 2 iNKT cells from *Med23*^{-/-} mice compared with WT controls (Fig. 1e). We have added this information in the revised manuscript (Page 8).

2. The inability to secrete cytokines despite activation-induced intracellular cytokine protein production is reminiscent of *Csf-2* deficient NKT cells: see Bezbradica et al. *Immunity* 2005. A commentary connecting this work to the current report could be useful.

We have investigated the transcriptional level of *Csf2* in stage 2 iNKT cells from WT and *Med23*^{-/-} mice. Interestingly, *Med23*-deficient stage 2 cells increased *Csf2* expression compared with WT controls (Supplementary Fig. 7c), indicating that *Csf-2* was not the reason for impaired iNKT cell function from *Med23*^{-/-} mice. We have added this information in the revised manuscript (Page 12).

3. Because *vav*-cre mediated *Med23* deletion in haematopoietic cells did not alter *Klrb1c* (NK1.1) expression, it was concluded that *Klrb1c* expression is not directly controlled by *Med23*. Nonetheless, it appears as though *Med23* deficient NKT cells do undergo terminal effector maturation but do not express the stimulatory and inhibitory NK receptors. So, the defect may be not so much stage 2 to stage 3 maturation but one of an inability to express stage 3/NKT1-specific surface markers. Over and above this defect lies the inability to secrete stage-specific effector cytokines and the deficiency in NKT17 subset.

We understand the reviewer's concern. We showed that not only iNKT cell surface makers but also effector functions were impaired in the absence of *Med23* in the previous version. To further compare the cell characteristics of WT stage 2, stage 3 cells and *Med23*-deficient stage 2 cells, we have analyzed their global gene expression profiles. As seen in Supplementary Figure 7b, principle component analysis showed that *Med23*-deficient stage 2 cells had a similar transcriptional space with WT stage 2 cells while WT stage 3 cells had a unique one compared to *Med23*-deficient stage 2 cells. Moreover, it is reported that NKT2 cells at stage 2 which do not secrete IL-4 in the steady state can develop to NKT1 cells at stage 3 in thymi and spleens³. Taken together, our data demonstrate that *Med23* deletion blocks iNKT cell development at stage 2 and impairs the terminal maturation of iNKT cells. We have added this information in the revised manuscript (Page 11).

4. The transcriptome of stage 2/NKT2, NKT17 and stage 3/NKT1 cells are well defined independently by the groups Brenner and Kronenberg (cite the latter that appeared in nat immunol). Therefore, it appears unusual to have redone the transcriptome of wild type stage 2 and stage 3 NKT cells without including Med23-deficient stage 2 cells thereby forfeiting the opportunity to define a Med23-controlled GRN the promotes stage 2 to stage 3 transition.

We agree with the reviewer. We had analyzed the gene expression profile of *Med23*-deficient stage 2 cells before we submitted our manuscript. Since we focused on the role of Med23 in the transition from stage 2 to stage 3 and transcriptome analysis of WT stage 2 and stage 3 cells was sufficient to answer our question, we did not display it in our initial submission. According to the reviewer's suggestion, we have added RNA-seq data of *Med23*-deficient stage 2 cells in our revised manuscript. Although WT and *Med23*-deficient stage 2 cells had similarly global transcriptomes (Supplementary Fig. 7a and b), pathway-enrichment analysis revealed that some genes had different expression levels in WT and *Med23*-deficient stage 2 cells (Supplementary Fig. 5). Interestingly, genes relating to the chromosome assembly exhibited significant changes indicating that Med23 may play important role in the gene expression of iNKT cells. We have added this information in the revised manuscript (Page 9, 11).

5. Is Med23 expression controlled by T-bet?

To answer the reviewer's question, we retrovirally expressed T-bet in thymic V α 14 iNKT cells and then measured the Med23 expression. As shown in Figure 1 (below), ectopic expression of T-bet in iNKT cells did not increase Med23 mRNAs indicating that T-bet did not control Med23 expression in the transcriptional level.

Figure 1. Ectopic expression of T-bet in iNKT cells does not affect Med23 transcription. Retrovirus was packaged by transfecting PlatE cell line with pMX-T-bet-IRES-GFP. The viral supernatants were collected 3 days after transfection. iNKT cells were sorted from WT V α 14 transgenic mice and culture with IL-7 (10 ng/ml). 24 h later, iNKT cells were infected by addition of virus in the presence of polybrene (8 μ g/ml). 24 h after the transduction, GFP⁺ iNKT cells were sorted. *Med23* mRNA levels in iNKT cells transduced to express T-bet or not were analyzed (V α 14 WT-vector, n = 4; V α 14 WT-T-bet, n = 5). The expression levels were normalized to *Gapdh* expression.

The data are presented as the mean \pm S.D. For the panels: N.S.: no significance. The data are combined from at least three independent experiments.

6. The manuscript dwells on the functional deficiency caused by the afore stated defects but overlooked the opportunity to dissect the functional capabilities of NKT2 cells.

We agree with the reviewer. It is reported that use of CD122 and CD4 can identify the iNKT functional subsets among the iNKT cells adequately⁴. We recognized the NKT2 cells as CD24^{low}TCR β ^{int}CD1d-PBS57⁺CD4⁺CD122⁻ and investigated how Med23 influence their function. According to the literature, we chose functional receptors CCR7, IL-6R α and cytokines IL-13, IL-4 which have high expression levels in NKT2 cells to investigate^{4,5}. One hour after administration with α -GalCer, we found that CCR7 and IL-6R α expression levels were comparable between WT and *Med23*-deficient NKT2 cells in spleens and livers. Intriguingly, NKT2 cells from *Med23*^{-/-} mice even had a slight increase in IL-6R α expression (Fig. 2a, below). Up on the activation of NKT2 cells, we also analyzed their cytokine secretion. Although splenic *Med23*-deficient NKT2 cells had a lower IL-4 secretion level than WT control, *Med23* deletion did not impair the IL-13 production by splenic and liver NKT2 cells. IL-4 production also maintained normal secretion level in livers (Fig. 2b-e, below). Moreover, *Med23* did not impact the frequency of NKT2 cells in stage 2 cells (Fig. 2g-i). Because these results indicate that *Med23* does not dramatically impact NKT2 cell function, we prefer not to include the functional assay of NKT2 cells in our revised manuscript. Certainly, we could add these data should the editor and the referee think.

Figure 2. Functional analysis of *Med23*-deficient NKT2 cells. (a) Representative histograms of surface CCR7 and IL-6R α in splenic and liver NKT2 cells from WT and *Med23*^{-/-} mice after 2 μ g of α -GalCer treatment for 1 h. (b and d) Production of IL-13 and IL-4 in WT and *Med23*^{-/-} NKT2 cells of the spleen (b) and liver (d) after in vivo stimulation for 1 h with 2 μ g of α -GalCer and then in vitro incubation for 2 h with brefeldin A. (c and e) The percentage of IL-13⁺ and IL-4⁺ cells among WT and *Med23*^{-/-} NKT2 cells in the spleens (c) and livers (e) after treatment by α -GalCer for 1 h and then administration with brefeldin A for 2 h (spleen, n = 6) (liver, IL-13⁺ cell frequency, n = 5; IL-4⁺ cell frequency, n = 6).

The data are presented as the mean \pm S.D. For all panels: *P < 0.05 by Student's t-test, N.S.: no significance. All data are representative of, or combined from, at least three independent experiments.

7. Reviews 1 and 11 are over a decade old; see Kumar et al. Front Immunol 2017 for a more recent review.

We thank the reviewer's suggestion. We have revised our references accordingly (Page 31).

8. Include Egawa et alii Immunity 2005, and Gapin et alii Nat Immunol 2002, along with the current ref. 10.

We take the advice. References have been updated in the revised manuscript (Page 32).

9. The manuscript is reasonably written considering the fact that it is written by non-native Authors. Nonetheless, there are several unusual usages of words and phrases. As well, there are grammatical errors, mostly of multiple tenses within the same sentence.

The English Language has been improved accordingly.

References

1. Matsuda JL, Gapin L, Sidobre S, Kieper WC, Tan JT, Ceredig R, *et al.* Homeostasis of V alpha 14i NKT cells. *Nat Immunol* 2002, **3**(10): 966-974.
2. Kennedy MK, Glaccum M, Brown SN, Butz EA, Viney JL, Embers M, *et al.* Reversible defects in natural killer and memory CD8 T cell lineages in interleukin 15-deficient mice. *J Exp Med* 2000, **191**(5): 771-780.
3. Lee YJ, Holzapfel KL, Zhu JF, Jameson SC, Hogquist KA. Steady-state production of IL-4 modulates immunity in mouse strains and is determined by lineage diversity of iNKT cells. *Nature Immunology* 2013, **14**(11): 1146-U1126.
4. Georgiev H, Ravens I, Benarafa C, Forster R, Bernhardt G. Distinct gene expression patterns correlate with developmental and functional traits of iNKT subsets. *Nat Commun* 2016, **7**: 13116.
5. Engel I, Seumois G, Chavez L, Samaniego-Castruita D, White B, Chawla A, *et al.* Innate-like functions of natural killer T cell subsets result from highly divergent gene programs. *Nat Immunol* 2016, **17**(6): 728-739.

Reviewer #2 (Remarks to the Author):

Although the work has been done in a thorough manner, a main issue with this report is the limited novelty of the findings. There is already a report on a defect in NKT cells from the knockout of a different subunit (Med1) of the mediator complex. (Yue et al PNAS 2011). These authors showed that the deficient mice exhibited defects in T-bet and CD122 expression, which would correlate with a decrease in stage 3 or NKT1 cells.

We thank the reviewer for his/her encouragement that our study has been done thoroughly.

We understand the reviewer's concern. It is reported that Med1 deficiency decrease rearrangement of TCR V α 14-J α 18 leading to impaired iNKT cell development from stage 0¹. Connected to our study, *Med23*^{-/-} DP thymocytes had comparable transcriptional levels of TCR V α 14-J α 18 compared to their WT counterparts (Supplementary Fig. 1c). In addition, loss of Med23 specifically blocked iNKT cell development at stage 2 and Med1 expression was not downregulated in *Med23*-deficient stage 2 cells (Fig. 3g). Moreover, our manuscript reports such a unique block during iNKT cell development, and thereafter provide a rare opportunity to study the terminal maturation of iNKT cells. We have added more discussion in the revised manuscript accordingly (Page 16).

A second problem is the use of markers such as NK1.1 and CD44 to separate stage 1, 2, 3 cells, designating maturation steps. This is acceptable in the thymus, although the consensus is that stage 2 NKT thymocytes are complex, because they contain precursors as well as real NKT2 cells. It has not been possible to reliably separate the precursors from NKT2 cells. In the periphery, however, it is uncertain if stage 1 and stage 2 immature NKT cells are exported from the thymus. Therefore, peripheral "stage 2" NKT cells likely are mostly NKT2 cells or an impure mixture rather than a true subset. Consistent with this, some of the stage 2 cells express high amounts of T-bet (Fig. 2) and therefore may not be immature at all.

This is a legitimate concern and is an issue for every study on iNKT cell development. Indeed, most peripheral NK1.1⁻ NKT cells have distinct function compared to thymic counterparts and homeostasis of iNKT cells in the periphery depends on the specific regulator^{2,3}. To clarify this issue, RAN-seq analysis of *Med23*-deficient stage 2 cells was added in the revised manuscript. Although *Med23* deficiency severely affected the transition from stage 2 to stage 3 (Fig. 1d and e), *Med23*-deficient stage 2 cells and WT stage 2 cells had comparable transcriptomes (Supplementary Fig. 7a and b), and the frequency of main functional subsets at stage 2 were not impaired in the thymi, spleens and livers from *Med23*^{-/-} mice (Fig. 2g-i). These results indicated that WT and *Med23*-

deficient stage 2 cells were similar in thymus and periphery even though the functional impairment were existed in *Med23*-deficient stage 2 cells. In addition, no literature has been reported that mature NK1.1⁻ NKT1 cells at stage 2 exist, and NKT2 cells which does not secret IL-4 in the steady state can develop to NKT1 cells in thymi and spleens⁴. All these studies suggest that thymic and peripheral stage 2 cells include developmental intermediates. We have discussed this issue in the revised manuscript (Page 7).

Detailed comments:

1. The authors should compare their RNA-Seq results to previous ones in which NKT1, NKT2 and NKT17 cells were analyzed. A PCA analysis of the data here combined with published data might be useful in showing how closely stage 2 NKT cells are to NKT2 cells that were selected using different surface markers.

We have compared our RNA-Seq results with the transcriptome analysis of NKT1, NKT2 and NKT17 cells (GEO: sequencing data, GSE69120)⁵ according to the reviewer's requirement. However, we cannot investigate the relationship of stage 2 cells and NKT2 cells correctly by principle component analysis because of different experimental methods (Fig. 3a, below). Alternatively, we used Pearson correlation analysis. As seen in Figure 3b (below), stage 2 cells seemed to be a mixture of NKT1, NKT2 and NKT17 cells and stage 3 cells were more similar to NKT1 cells.

Figure 3. Gene expression analysis of iNKT cells at different stages and functional subsets. (a) Gene expression profiles of stage 2 and stage 3 cells from WT mice and NKT1 cells, NKT2 cells, NKT17 cells from published data were transformed by principle component analysis. (b) Pearson correlation analysis of stage 2 and stage 3 cells from WT mice and NKT1 cells, NKT2 cells, NKT17 cells from published data.

All data are combined from at least three independent experiments.

2. Differences in the experimental protocols in Figs. 5b and 5d need to be more clearly explained in the text and figure legends, considering that the percent of NKT cells producing cytokine is different in the two panels.

We thank the reviewer's suggestion and we have added a clear description of how experiments were performed in the revision according to the reviewer's suggestion (Page 12, 41).

3. It has been shown that localization of NKT cell subsets within the tissues influences their cytokine responses (YJ Lee et al, Immunity 2015). For example, intravenous injection of α GalCer activated NKT1 cells in spleen (and liver) which locate in red pulp whereas NKT2 cells in the T cell zone were less responsive. The data in Fig. 5 are consistent with this idea, meaning a higher proportion of NKT2 cells, rather than a general defect in the ability of the NKT cells to be activated by antigen. Do the *Med23* deficient NKT cells have a defect in TCR signaling as well? Figure 5f-i suggest this could be the case, although the differences are not great and NKT2 cells were not adequately purified. What is their response to PMA+ionomycin or IL12+IL18 stimulation?

To determinate that if the different locations of WT and *Med23*^{-/-} iNKT cells resulting in their different cytokine responses, we have studied cytokine production of vascular localized iNKT cells from WT and *Med23*^{-/-} mice. To label vascular leukocytes, mice were injected intravenously with anti-CD45 antibody for 3 minutes before they were sacrificed. As seen in Figure 4a-d (below), Splenic *Med23*-deficient stage 2 cells produced the lowest amounts of cytokines and WT stage 3 cells had the highest levels of IFN- γ and IL-4 secretion. In livers, *Med23*-deficient stage 2 cells also impaired their IFN- γ production compared with WT stage 3 cells. These results demonstrate that *Med23* is an intrinsic regulator of iNKT cell function.

Regarding to the reviewer's second point, we are not able to investigate the defect in TCR signaling. Because iNKT cells sorted by TCR β and CD1d-PBS57 staining may stimulate their TCR signaling during cell sorting and culture. Alternatively, we have stimulated total splenic and liver leukocytes from WT and *Med23*^{-/-} mice by PMA and ionomycin. As shown in Figure 5a-d (below), *Med23*-deficient stage 2 cells had a similar IFN- γ production with WT stage 2 cells in spleens and livers and increased the IL-4 secretion compared with WT stage 2 and stage 3 cells in spleens. These results remained somewhat uncertainty since some side effects might overcome the functional defect in *Med23*-deficient stage 2 cells after PMA and ionomycin stimulation.

Figure 4. Med23 regulates cytokine secretion of vascular localized iNKT cells. (a and c) Production of IFN- γ and IL-4 in vascular localized (IV⁺) WT stage 2 and stage 3 cells and *Med23*^{-/-} stage 2 cells of the spleen (a) and liver (c) after in vivo stimulation for 1 h with 2 μ g of α -GalCer and then in vitro incubation for 2 h with brefeldin A. (b and d) The percentage of IFN- γ ⁺ and IL-4⁺ cells among vascular localized WT stage 2 and stage 3 cells and *Med23*^{-/-} stage 2 cells in the spleens (b) and livers (d) after α -GalCer administration and then in vitro incubation for 2 h in the presence of brefeldin A (spleen, IFN- γ ⁺ cell frequency, n = 6; IL-4⁺ cell frequency, IV⁺ WT stage 2 and stage 3, n = 6, IV⁺ KO stage 2, n = 5) (liver, n = 6).

The data are presented as the mean \pm S.D. For all panels: *P < 0.05; **P < 0.01; ****P < 0.0001 by Student's t-test, N.S.: no significance. All data are representative of, or combined from, at least three independent experiments.

Figure 5. Loss of Med23 does not impair cytokine secretion of iNKT cells after the PMA and ionomycin stimulation. (a and c) Production of IFN- γ and IL-4 in WT stage 2 and stage 3 cells and *Med23*^{-/-} stage 2 cells of the spleen (a) and liver (c) after stimulation with PMA (50 ng/ml) and ionomycin (1 μ g/ml) in the presence of brefeldin A for 4 h. (b and d) The percentage of IFN- γ ⁺ and IL-4⁺ cells among WT stage 2 and stage 3 cells and *Med23*^{-/-} stage 2 cells in the spleens (b) and livers (d) after treatment by PMA and ionomycin in the presence of brefeldin A for 4 h (spleen, IFN- γ ⁺ cell frequency, n = 5; IL-4⁺ cell frequency, n = 8) (liver, IFN- γ ⁺ cell frequency, n = 5; IL-4⁺ cell frequency, WT stage 2, n = 7, KO stage 2 and WT stage 3, n = 8).

The data are presented as the mean \pm S.D. For all panels: **P < 0.01; ***P < 0.001 by Student's t-test, N.S.: no significance. All data are representative of, or combined from, at least three independent experiments.

4. The authors should note that the data in Fig. 6 do not establish a causative link between cell recruitment and CCL5 expression by NKT cells. CCL5KO or knockdown NKT cells would be needed to assess this in a transfer system.

We thank the reviewer's suggestion which significantly improves our study. To address this question, we initially tried to transfer CFSE-labeled liver iNKT cells into *Ja18*^{-/-} mice (iNKT cell-deficient) and investigated CFSE⁺ iNKT cells in liver (day 1, day 3 and day 6). Only few CFSE⁺ cells were detected in the liver. The same situation has been reported in the published data⁶. Alternatively, we injected WT mice a CCL5 antagonist Met-RANTES or control PBS before α -GalCer stimulation and measured neutrophil cellularity indicating the recruitment capacity of iNKT cells. As shown in Supplementary Figure 8, PBS-treated mice significantly increased the cell number of neutrophils in livers after activation of iNKT cells. But Met-RANTES-treated mice only increased the cell number slightly contrasted to the PBS controls. Our results reveal that Med23 influencing CCL5 secretion of iNKT cells may be one of the reasons for regulation of iNKT cell recruitment capacity. We have added this information in the revised manuscript (Page 14).

5. Some of the data have high error bars such as thymus spleen cell numbers in Fig. 1 b or thymus cells number in figure 2f. A scatter gram showing the values from individual mice would cause some clutter, but would be more revealing than a bar graph. Was the appropriate statistical test done?

We have changed our statistical analyses according to the reviewer's suggestion.

6. On page 9 a reference to fig. 4d-I should read: fig. 2d-i.

Sorry for our mistake. We have corrected this in the revised manuscript (Page 7).

References

1. Yue X, Izcue A, Borggrefe T. Essential role of Mediator subunit Med1 in invariant natural killer T-cell development. *Proc Natl Acad Sci U S A* 2011, **108**(41): 17105-17110.
2. McNab FW, Pellicci DG, Field K, Besra G, Smyth MJ, Godfrey DI, *et al.* Peripheral NK1.1 NKT cells are mature and functionally distinct from their thymic counterparts. *J Immunol* 2007, **179**(10): 6630-6637.
3. Yu SH, Cantorna MT. The vitamin D receptor is required for iNKT cell development. *P Natl Acad Sci USA* 2008, **105**(13): 5207-5212.
4. Lee YJ, Holzapfel KL, Zhu JF, Jameson SC, Hogquist KA. Steady-state production of IL-4 modulates immunity in mouse strains and is determined by lineage diversity of iNKT cells. *Nature Immunology* 2013, **14**(11): 1146-U1126.
5. Georgiev H, Ravens I, Benarafa C, Forster R, Bernhardt G. Distinct gene expression patterns correlate with developmental and functional traits of iNKT subsets. *Nat Commun* 2016, **7**: 13116.
6. Crowe NY, Coquet JM, Berzins SP, Kyparissoudis K, Keating R, Pellicci DG, *et al.* Differential antitumor immunity mediated by NKT cell subsets in vivo. *J Exp Med* 2005, **202**(9): 1279-1288.

REVIEWERS' COMMENTS:

Reviewer #1 (Remarks to the Author):

The Authors have carefully addressed the concerns raised in the previous review and have returned a properly revised manuscript. Nonetheless, written language issues remain.

Reviewer #2 (Remarks to the Author):

The authors have improved the manuscript with transcriptomic and other additional data. Their principal claim that the Med23 ko provides a unique block in NKT cell differentiation seems to be a little exaggerated. It is true that the Med1 ko has some impairment (2-fold) in stage 0 NKT cells that was not found in the Med23 ko, and the phenotype in the Med1 ko is a little more severe. However, the two strains seem rather similar in selective or most severe stage 3 deficiency. The authors should discuss their findings relative to the Med1 ko and to other gene deficient mouse strains (Hobit, vitamin D receptor, others?) also reported to have a somewhat similar NKT cell stage 3 deficiency.

REVIEWERS' COMMENTS:

Reviewer #1 (Remarks to the Author):

The Authors have carefully addressed the concerns raised in the previous review and have returned a properly revised manuscript. Nonetheless, written language issues remain.

We thank the reviewer for his/her positive comments and the written language has been improved by the Nature Research Editing Service.

Reviewer #2 (Remarks to the Author):

The authors have improved the manuscript with transcriptomic and other additional data. Their principal claim that the Med23 ko provides a unique block in NKT cell differentiation seems to be a little exaggerated. It is true that the Med1 ko has some impairment (2-fold) in stage 0 NKT cells that was not found in the Med23 ko, and the phenotype in the Med1 ko is a little more severe. However, the two strains seem rather similar in selective or most severe stage 3 deficiency. The authors should discuss their findings relative to the Med1 ko and to other gene deficient mouse strains (Hobit, vitamin D receptor, others?) also reported to have a somewhat similar NKT cell stage 3 deficiency.

We take the advice. We have toned down our conclusions in the revised manuscript. In addition, we have discussed our study relative to other findings of similar deficiencies of stage 3 iNKT cells in gene-deficient mouse strains in the revised manuscript (Page 18).